# DYNAMIC MASK ATTENTION: END-TO-END TRAINABLE CONTENT-AWARE SPARSE ATTENTION

## ABSTRACT

Self-attention's computational cost, which scales quadratically with sequence length, creates a fundamental bottleneck for long-context modeling in LLMs, limiting applications such as document understanding, multi-turn reasoning, and code generation. Sparse attention has been proposed to mitigate this issue. Early *content-agnostic* designs such as sliding-window and block-sparse attention reduce computational complexity based on fixed patterns. However, their static structure often overlook important long-range dependencies and lack adaptivity to diverse query contexts. Recent *content-aware* methods improve adaptivity by conditioning attention sparsity on token representations, but they typically rely on hard binary masks or heuristic key-value selection, introducing runtime overhead and hindering fully differentiability. We propose *Dynamic Mask Attention (DMA)*, a trainable content-aware sparse attention mechanism with head-wise specialization. DMA *dynamically* generates content-driven dynamic masks with continuous importance weights based on value representations, enabling both expressiveness and full differentiability. We theoretically prove that masked entries are mathematically equivalent to zero in both the forward and backward passes, thereby ensuring unbiased gradients. Furthermore, we developed efficient CUDA kernels with block-skipping for practical acceleration. Extensive experiments demonstrate that DMA consistently outperforms state-of-the-art sparse attention baselines across pre-training and downstream tasks, reducing perplexity, improving accuracy, and delivering substantial long-sequence speedups of up to $10\times$. Our implementation is released in an anonymous repository and has been integrated into widely used frameworks, supporting adoption in large-scale LLM training. https://github.com/dma-anonymity/dma

## 1 INTRODUCTION

Self-attention's (Vaswani et al., 2017) quadratic computation complexity is a core bottleneck (Zaheer et al., 2020) for practical long-context modeling (Snell et al., 2024) in LLMs. This limitation becomes critical in scenarios such as long-document understanding (Park et al., 2023; DeepMind, 2025), multi-turn reasoning (HuggingFace, 2025; Guo et al., 2025), and codebase generation (Zhang et al., 2024; Team, 2025), where both precise long-range recall (Liu et al., 2024c) and efficient inference (Kwon et al., 2023) are essential.

To mitigate these challenges, a broad class of methods introduces *sparse attention*, which reduces the quadratic cost by computing only a subset of query–key interactions. Early designs are **content-agnostic sparse attention**, relying on pre-defined positional patterns such as sliding-window, strided, or block/global hybrids like Longformer (Beltagy et al., 2020) and BigBird (Zaheer et al., 2020). While these patterns reduce computational costs, they inevitably miss semantically relevant but distant tokens and apply the same mask to all inputs, making them poorly suited for query-dependent long-context modeling (Liu et al., 2024a). To overcome these limitations, recent work introduces **content-aware sparse attention**, which determines important tokens based on the input itself. Some methods first compute importance surrogates and then build a sparse mask (e.g., DAM (Zhang et al., 2025), Quest (Tang et al., 2024)), which introduces a two-stage design and extra runtime overhead. Others dynamically select or prune the KV cache (e.g., H2O, SnapKV, InfLLM), which reduces memory traffic but relies on non-trainable heuristics and thus cannot be optimized end-to-end (Li et al., 2024; Zhang et al., 2023; Xiao et al., 2024a). Recent approaches

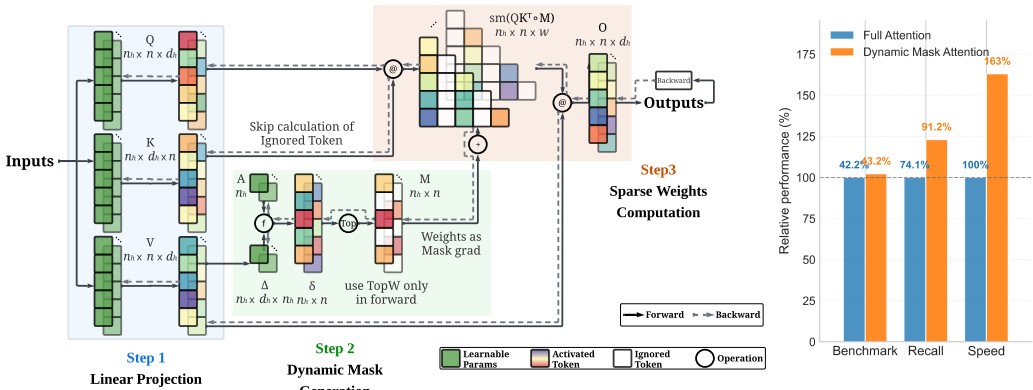

Figure 1: **Overall workflow of Dynamic Mask Attention (DMA)**. **Left:** The overall workflow of DMA. Step 1 projects inputs into $Q$, $K$, and $V$. Step 2 generates content-aware dynamic masks with continuous importance scores, skipping irrelevant tokens. Step 3 applies sparse weight computation with gradient flow, ensuring both efficiency and differentiability. **Right:** Relative performance comparison between full attention and DMA across representative benchmarks. DMA achieves higher recall and significantly faster speed, while maintaining competitive accuracy.

like MoBA (Lu et al., 2025) and FlexPrefill (Lai et al., 2025) attempt to improve adaptability through block-level routing or statistical metrics. However, they either operate at a coarse granularity or rely on hand-crafted criteria (e.g., Jensen-Shannon divergence), limiting their ability to learn optimal fine-grained sparsity patterns directly from data. Native Sparse Attention (NSA) (Yuan et al., 2025) makes progress by introducing a trainable sparse attention mechanism, but its multi-branch design still incorporates fixed patterns such as local or strided attention. As a result, although trainable, some branches still rely on pre-defined patterns, preventing fully adaptive optimization from semantic content. More sparse attention methods are introduced in Appendix B.

In summary, existing methods remain limited by two key issues: (i) hard binary masks that only keep or discard tokens, failing to represent the continuous spectrum of token importance; and (ii) discrete gating that prevents gradient flow through the mask generation process, blocking end-to-end optimization with the model. These limitations motivate the need for an **end-to-end trainable** sparse attention mechanism that can **adaptively learn content-aware sparsity** in long texts.

**Our Method.** To address these challenges, we introduce **Dynamic Mask Attention (DMA)**, *an end-to-end trainable and content-aware sparse attention mechanism with head-wise specialization*. As shown in Figure 1, DMA generates content-aware, dynamic masks that assign continuous importance weights to tokens, enabling end-to-end optimization alongside the modeling objective for long texts. Specifically, our design introduces two key ideas: (i) a *soft-gating* mask that replaces binary keep-or-drop decisions with continuous values, thereby capturing fine-grained variation in token importance; and (ii) a *differentiable content-aware mask generation algorithm* based on value representations, which allows gradients to flow through the mask and enables adaptive learning of optimal sparsity patterns. Crucially, both the introduced dynamic mask and the sparse weight computation are designed to be fully differentiable: the preserved attention paths yield gradients that are strictly identical to those of full attention, ensuring that gradient flow reaches all inputs and parameters without being blocked by discrete operations. This guarantees stable end-to-end training and aligns with our goal of retaining critical dependencies while suppressing redundant computation.

**Contributions.** Our work makes the following contributions:

- We introduce Dynamic Mask Attention (DMA), a new sparse attention mechanism for long-context transformers that generates content-aware masks from value representations and supports head-wise specialization.

- We develop efficient CUDA kernels with block-skipping to realize practical speedups. Our implementation has been integrated into HuggingFace Transformers, is being integrated into Py-Torch's SDPA backend, and has been adopted by Company X (details omitted for anonymity).

- DMA consistently outperforms state-of-the-art methods. It reduces pre-training perplexity by 2–3.5% over NSA (Yuan et al., 2025) across 80M–1.7B models; delivers 38–87% faster inference

at sequence lengths up to 8192 tokens; achieves up to 487% speedup over NSA on multi-query associative recall; improves zero-shot accuracy by 5% at 1.7B scale; and boosts needle-in-a-haystack retrieval by 14.3% compared to NSA.

## 2 METHODOLOGY

### 2.1 DYNAMIC MASK GENERATION

**Variable Initialization.** Assume that the multi-head attention input is $Q \in \mathbb{R}^{n_h \times L_q \times d_h}, K, V \in \mathbb{R}^{n_h \times L_k \times d_h}$, where $n_h$ is the number of heads, $L_q, L_k$ is the query/key sequence length, and $d_h$ is the intra-head dimension. We introduce a sampling stride weight $\Delta \in \mathbb{R}^{d_h}$ and a head-by-head gating coefficient $A \in \mathbb{R}^{n_h}$; let $\tau(\cdot)$be the non-negative range activation function. The method is divided into two stages: first, a sparse mask is generated using the contents of the $V$ cache, and then attention weights are calculated on the unmasked region.

$$\delta_{h,k} = \exp\Big( \tau\big(\langle V_{h,k,:}, \Delta \rangle\big) \cdot A_h \Big), \delta \in \mathbb{R}^{n_h \times L_k} \tag{1}$$

The core of the first step is to generate a learnable importance score based on the content. In Equation equation 1, $V$ is the input Value tensor, representing the content information of each token, and $\delta$ is the learnable projection weight. First, a linear projection of $V$ is performed via tensor contraction $V \cdot \delta$, converting each token's $d_h$-dimensional content vector into a scalar feature that serves as a preliminary estimate of its importance. Subsequently, a non-negative activation function $\tau(\cdot)$ ensures that the score is non-negative to avoid signal suppression. Next, a per-head gating coefficient $A$ scales the importance score for each head, allowing the model to learn the sparsity of different heads. Finally, an exponential function $exp(\cdot)$ maps the final score to a positive space and increases the gap between high and low scores, making important tokens more prominent. This entire process does not rely on the query or key, but rather dynamically calculates a score for each token position based directly on the content of the Value vector itself.

**Mask Generation.** According to the score $\delta$ obtained in the previous step, the most important $w$ tokens are selected for each head, and a mask $M$ is generated.

$$M_{h,k}^K = \begin{cases} \phi(\delta_{h,k}), & k \in \mathcal{K}_h \\ -\infty, & \text{otherwise} \end{cases}, \text{ where } \mathcal{K}_h = \text{TopK}(\delta_{h,:}, w) \tag{2}$$

This step performs a sparse selection, converting the continuous score from the previous step into a "soft" sparse decision. The TopK($\delta$, w) operation in the formula finds the top w values in the score tensor $\delta$ and returns their index set $K_h$. Next, for each key position index $k$, it is determined whether it is in the set of the most important $w$ positions. If it is, its original relevance score is retained through $\Phi(\delta_{h,k})$; if not, its value is set to $-\infty$. In this way, for selected tokens, we retain their specific scores to ensure that the gradient can be propagated back; for unselected tokens, we mark them with $-\infty$ so that their probabilities approach 0 in the subsequent softmax.

**Mask Broadcasting.** The one-dimensional content mask $M^K$ is expanded to two dimensions, and causality is incorporated to ensure that the model does not see future information.

$$M_{h,q,k} = T_{q,k} \cdot M_{h,k}^K \tag{3}$$

This step ensures the model's autoregressive properties. In the formula, $M_{h,k}^K$ is the content-based one-dimensional mask generated in the previous step, while $T_{q,k}$ is the causality matrix (when k $\leq$ q, 0 otherwise). By combining these two, the resulting two-dimensional mask $M_{h,q,k}$ ensures that a token is retained for subsequent calculations only if it satisfies both the "content-important" and "current or past" conditions.

**Additive Masked Attention.** Calculate the dot product of Query and Key and apply the resulting mask $M$ in an additive manner.

$$S_{h,q,k} = \frac{\langle Q_{h,q,:}, K_{h,k,:} \rangle}{\sqrt{d_h}} + M_{h,q,k} \tag{4}$$

This step is the core of enforcing the sparsity constraint. First, a standard scaled dot-product attention score is computed via $<Q, K>/sqrt(d_h)$. Then, the 2D mask $M_{h,q,k}$ generated in the previous

step is applied additively. For positions marked as $-\infty$, their final score $S$ also becomes $-\infty$. For positions that are retained, their original importance scores are added to the attention scores as a learnable bias. This allows the parameters ($\delta$, $A$) of the entire mask generation process to be optimized end-to-end via gradient descent.

**Sparse Attention Calculation.** The masked score $S$ is converted into a probability distribution using the Softmax function.

$$P_{h,q,k} = \text{Softmax}_k(S_{h,q,k}) \tag{5}$$

This step converts the scores into valid probabilities. $S_{h,q,k}$ in the formula is the attention score calculated in the previous step, which incorporates sparse information. The Softmax function normalizes it along the key dimension $k$. Since the scores of unselected positions are $-\infty$, and the result of $exp(-\infty)$ approaches 0, the normalized probability weights for these positions are almost zero. Ultimately, the attention weight P will be concentrated only on the selected $w$ tokens, forming a sparse attention distribution.

**Output Aggregation.** Finally, the sparse attention weight $P$ is used to perform weighted summation on the Value vector to obtain the final output.

$$O_{h,q,:} = \sum_{k=1}^{L_k} P_{h,q,k} V_{h,k,:} \tag{6}$$

This step is information aggregation. Like the standard attention mechanism, it performs a weighted summation of the original Value vector $V$ based on the sparse attention weights $P$ calculated in the previous step. However, because $P$ is sparse, this summation operation actually only computes a small number $w$ of Value vectors, ignoring the vast majority of items with zero weights. This speeds up the computation and yields the final output O.

## 2.2 How To Make a Dynamic Mask Trainable?

The goal of the backward pass is to compute the gradients of the loss with respect to the inputs $Q$, $K$, $V$, and the learnable parameters $M$. To save memory, the algorithm does not load the attention score matrix $S$ and probability matrix $P$ from the forward pass. Instead, it recomputes them on the fly during the backward pass based on the current block's $Q$, $K$, and $M$.

$$S_{block} = (Q_{block}@K_{block}^T) + M_{block} \tag{7}$$

$$P_{block} = softmax(S_{block}) \tag{8}$$

When processing each pair of $Q_{block}$ and $K_{block}$, the algorithm first recomputes the attention scores $S$ between them, where $M_{block}$ corresponds to the mask tensor. It then applies the softmax function to obtain the sparse attention probabilities $P$.

Gradient calculation follows the chain rule, propagating backward layer by layer from the model's output $O$. We use $d$ to denote the gradient; for example, $dO$ is the gradient of the loss with respect to the output $O$.

First, the gradient with respect to the Value matrix $V$ is calculated. The final step of the forward pass is $O = P @ V$. According to matrix calculus rules, the gradient of the loss with respect to $V$, denoted as $dV$, is:

$$dV = P^T@dO \tag{9}$$

This formula calculates how the upstream gradient $dO$ is distributed back to the Value vectors $V$ through the attention matrix $P$. Intuitively, a Value vector's contribution to the final output is determined by its corresponding attention weight in $P$, so it receives the gradient according to that same weight during backpropagation.

Next, the gradient with respect to the attention probability matrix $P$, denoted as $dP$, is calculated:

$$dP = dO@V^T \tag{10}$$

This step calculates how the upstream gradient $dO$ is propagated to the attention probabilities $P$ through the Value matrix $V$.

This is the most critical step for demonstrating the mask's differentiability. In the forward pass, the attention score is $S = (Q@K^T) + M$. According to the chain rule, the gradient with respect to the mask $M$, $dM$, is equal to the gradient with respect to the attention scores $S$, $dS$.

$$dM = dS = (dP - (P * dP).sum(axis)) * P \qquad (11)$$

The core of this formula is to calculate the gradient of the mask $M$, $dM$. This is achieved by backpropagating the gradient $dP$ through the softmax function, and the result is numerically equal to $dS$. The calculated $dM$ contains the crucial gradient information that will be further propagated to the parameters that generate the mask ($\delta$ and $A$), enabling end-to-end training.

Finally, we use the just-calculated mask gradient $dM$ to compute the gradients for $Q$ and $K$. Since $dM$ is equal to $dS$, it can be applied to the $Q@K^T$ term.

$$dQ = dM@K \qquad (12)$$

$$dK = dM^T@Q \qquad (13)$$

Because the forward pass includes the term $S = (Q@K^T) + M$, the upstream gradient $dM$ is multiplied by $K$ and $Q$, respectively, to obtain $dQ$ and $dK$. This completes the gradient propagation from the mask's computational node to $Q$ and $K$.

Due to space limitations, we also include our complete proof in Appendix D.

## 2.3 EFFICIENT AUTOREGRESSIVE DECODING WITH DYNAMIC KV CACHE

The previous subsections describe how Dynamic Mask Attention constructs a content-aware sparse mask for a single forward pass. During autoregressive decoding, however, the key–value cache grows linearly with the number of generated tokens $t$. Naively recomputing a global top-$w$ selection over the entire prefix at every step would incur $\mathcal{O}(t)$ computational overhead per query, where $t$ is the number of generated tokens. To keep both the sparsity pattern and the computational cost stable as $t$ increases, we design an incremental value-driven KV selection scheme.

**Global value-driven KV subset.** For each attention head $h$, we maintain a persistent set of key indices $I_h^{(t)}$ whose size is bounded by $w_{\mathrm{kv}}$. This set stores the positions of keys/values that are kept resident in high-bandwidth memory (HBM). Let $\delta_{h,i}$ denote the importance score of token $i$ for head $h$. When the $(t+1)$-th token is appended, we compute its score $\delta_{h,t+1}$ and update $I_h^{(t)}$ via an incremental Top-$w_{\mathrm{kv}}$ rule:

$$I_h^{(t+1)} = \begin{cases} I_h^{(t)} \cup \{t+1\}, & \text{if } |I_h^{(t)}| < w_{\mathrm{kv}}, \\ \left(I_h^{(t)} \setminus \{i^*\}\right) \cup \{t+1\}, & \text{if } \delta_{h,t+1} > \min_{i \in I_h^{(t)}} \delta_{h,i}, \ i^* = \arg\min_{i \in I_h^{(t)}} \delta_{h,i}, \\ I_h^{(t)}, & \text{otherwise.} \end{cases} \qquad (14)$$

Intuitively, if the set is not yet full, we simply insert the new position $t+1$. Once $|I_h^{(t)}| = w_{\mathrm{kv}}$, the new token replaces the currently least important element in $I_h^{(t)}$ only if its score is larger; otherwise, the global set remains unchanged. The keys and values removed from $I_h^{(t)}$ are swapped out from HBM to slower memory (e.g., host DRAM), while the active KV tensors used in the attention computation for head $h$ at step $t$ are

$$K_h^{\mathrm{act},(t)} = [k_{h,i}]_{i \in I_h^{(t)}} \in \mathbb{R}^{w_{\mathrm{kv}} \times d_h}, \qquad V_h^{\mathrm{act},(t)} = [v_{h,i}]_{i \in I_h^{(t)}} \in \mathbb{R}^{w_{\mathrm{kv}} \times d_h}. \qquad (15)$$

This guarantees that the resident KV cache per head occupies only $\mathcal{O}(w_{\mathrm{kv}} d_h)$ memory, or $\mathcal{O}(n_h w_{\mathrm{kv}} d_h)$ in total.

**Query-dependent dynamic slots.** A purely global Top-$w_{\mathrm{kv}}$ subset is query-independent and may miss tokens that are temporarily less important globally but highly relevant to the current query $q_{h,t}$. To restore query adaptivity with small overhead, we introduce at most $w_q$ *query-dependent dynamic slots* for each decoding step. We first construct a candidate index set $C_t$ that contains a small number of potentially useful tokens, such as the most recent window of tokens, the few tokens

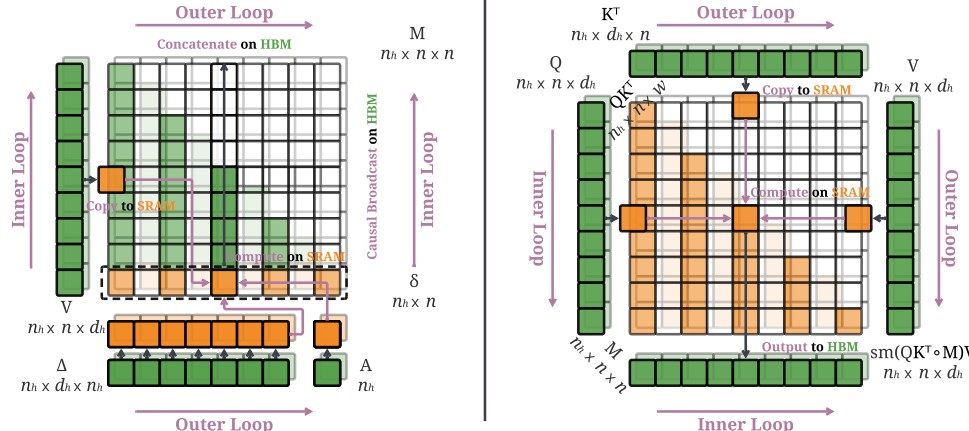

Figure 2: **Dynamic Mask Attention Architecture**. **Left**: Step 1: Dynamic Mask Generation. The mask computation part of the dynamic mask attention. **Right**: Step 2: Sparse Weights Computation. The weight computation part of the dynamic mask attention.

that were recently evicted from $I_h^{(t)}$, or the latest incremental batch of appended tokens. For each head $h$ and each $i \in C_t$, we compute a relevance score using the standard scaled dot product:

$$r_{h,i,t} = \frac{\langle q_{h,t}, k_{h,i} \rangle}{\sqrt{d_h}}, \qquad i \in C_t. \tag{16}$$

This query-aware decoding step follows the lightweight on-the-fly candidate re-ranking strategy proposed in NOSA (Huang et al., 2025), which we adapt to the value-driven DMA cache. We then select from $C_t$ the positions with the highest scores that are not already in the global set:

$$Q_h^{(t)} = \text{Top}_{w_q} \left\{ r_{h,i,t} \, \middle| \, i \in C_t \setminus I_h^{(t)} \right\}. \tag{17}$$

The effective attention index set for query $q_{h,t}$ is given by

$$\mathcal{A}_h^{(t)} = I_h^{(t)} \cup Q_h^{(t)}, \qquad |\mathcal{A}_h^{(t)}| \leq w_{\text{kv}} + w_q. \tag{18}$$

Attention scores and value aggregation are only computed on indices in $\mathcal{A}_h^{(t)}$. As a result, the per-step complexity of decoding is reduced from $\mathcal{O}(n_h t d_h)$ to $\mathcal{O}\big(n_h(w_{\text{kv}} + w_q)d_h\big)$, while the KV cache stored in HBM remains at $\mathcal{O}(n_h w_{\text{kv}} d_h)$. The dynamic slots $Q_h^{(t)}$ are recomputed for each query and do not need to be stored persistently. In practice, we maintain a small min-heap per head to implement the incremental update in Eq. equation 14.

## 2.4 THE KERNEL DESIGN OF DMA

In kernel design, the entire computation flow is shown in the left of Figure 9, in the outer loop, the stride weight $\Delta$ and gate weight $A$ are loaded into high-speed SRAM, and in the inner loop, the zero-order hold method is used to loop through the $V$ blocks loaded into SRAM, sampling from it to generate content-aware $K$ masks. These masks are then causally broadcast to the length of $Q$ in High Bandwidth Memory (HBM) to avoid quadratic complexity memory usage. Finally, in the outer loop, all mask blocks are concatenated to form the final content-aware sparse dynamic mask. Please refer to Appendix D for details.

## 3 EXPERIMENTS

### 3.1 EXPERIMENTAL SETTINGS

**Baselines.** We evaluate DMA through five lenses: (1) Scaling Perplexity to assess language-modeling quality across parameter sizes, (2) Multi-Query Associative Recall to measure long-sequence retrieval accuracy and report inference speed, (3) Implementation Comparison to benchmark DMA against other efficient variant implementations from three dimensions, (4) Downstream

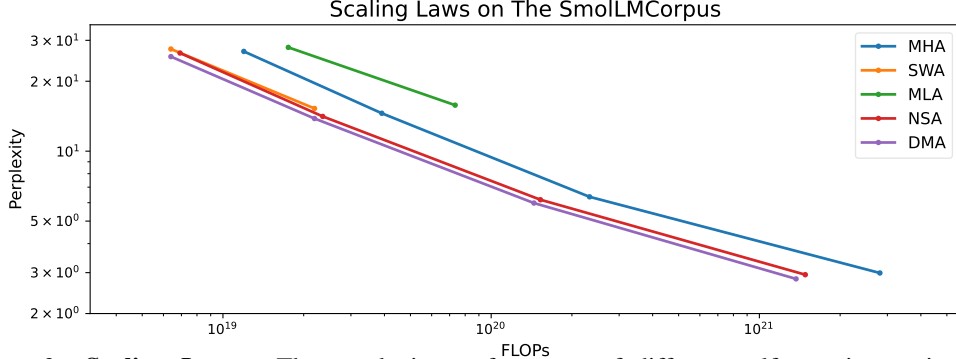

Figure 3: **Scaling Laws**. The perplexity performance of different self-attention variants on SmolLMCorpus at different parameter scales. For suboptimal variants like SWA and MLA, we omit them for clarity. Compared to other variants, our Dynamic Mask Attention has a Pareto advantage in performance.

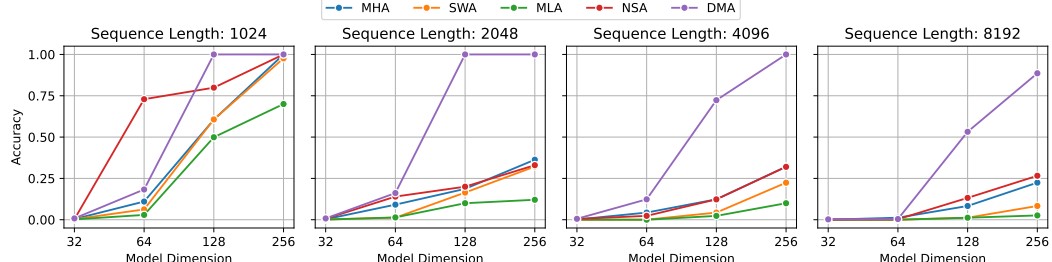

Figure 4: **Multi-Query Associative Recall**. This is a more challenging version of the original multi-query associative recall task (Arora et al., 2024), which includes longer sequence lengths and smaller model dimensions. Dynamic Mask Attention maintains good performance in most cases.

Benchmarks contrasting DMA with the Full-Attention baseline and advanced sparse attention methods, and (5) Extrapolated Content Retrieval to evaluate length extrapolation and long-context retrieval.

**Methods.** In Section 3.2, we compare five attention variants: MHA (Vaswani et al., 2017), SWA (Fu et al., 2025), MLA (Liu et al., 2024a), NSA, and our DMA. In Section 3.3, we benchmark the kernel-level acceleration of DMA against other variant efficient implementations across different dimensions. For Section 3.4, in addition to MHA, NSA, and DMA, we also include H2O (Zhang et al., 2023), InfLLM (Xiao et al., 2024a), Quest (Tang et al., 2024), DAM (Zhang et al., 2025), and Exact-Top. Finally, in Extrapolated Content Retrieval, we compare MHA, NSA, and DMA to assess long-context retrieval and length extrapolation.

**Training Settings.** All experiments were conducted using the open-source PyTorch images (NVIDIA, 2022) and the Transformers framework (Wolf et al., 2020). We use SmolLM-Corpus (Ben Allal et al., 2024) a curated, fully deduplicated mixture of high-quality Web, code, math, academic, and textbook sources emphasizing permissive licensing and low noise, as training data. For evaluation frameworks, we utilized the LM evaluation harness (Gao et al., 2021) from EleutherAI for perplexity tasks, and the lighteval (Fourrier et al., 2023) from HuggingFace for downstream tasks.

### 3.2 VARIANTS COMPARISON

**Scaling Perplexity.** Figure 3 compares the perplexity of our Dynamic Mask Attention against several variants, including MHA (Vaswani et al., 2017), SWA (Beltagy et al., 2020), MLA (Liu et al., 2024a), and NSA (Yuan et al., 2025). Across model sizes from 80M to 1.7B trained on the SmolLMCorpus dataset (Ben Allal et al., 2024), DMA consistently achieves the best performance. We attribute this advantage to its ability to adaptively focus on key information, thereby mitigating the "lost-in-the-middle" problem (Liu et al., 2024c). Experimental configurations are detailed in Table 3.

---

[0]The implementation code for NSA is available at `https://github.com/lucidrains/native-sparse-attention-pytorch`.

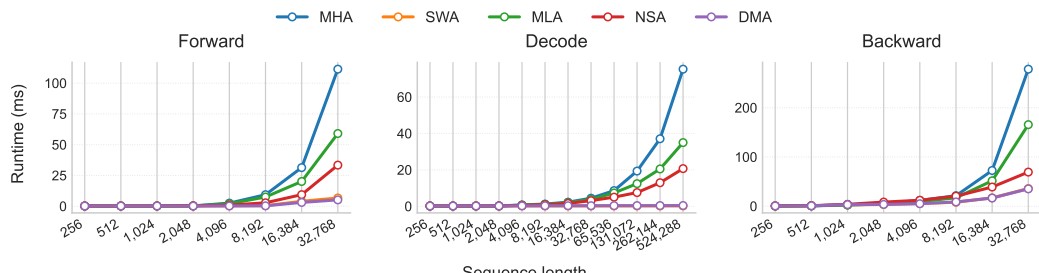

Figure 5: **Kernel Performance**. A performance comparison of efficient implementation kernels of different attention variants on an A100 GPU. Our DMA achieves significant acceleration in forward propagation, decoding, and backward propagation, maintaining the same efficiency level as SWA.

**Associative Recall.** To test long-sequence information retrieval, we used a more challenging multi-query associative recall task (Arora et al., 2024), which included 512 key-value pairs and replaced irrelevant context with random tokens to increase difficulty. To ensure a fair comparison, we not only set a uniform window size of 512 during evaluation, but also trained all models for 100 epochs on a dataset containing 250,000 training samples. As shown in Figure 4, DMA excels across all sequence lengths, demonstrating its ability to effectively focus on relevant information while ignoring noise.

### 3.3 IMPLEMENTATION COMPARISON

**Kernel Acceleration.** To analyze the performance of Dynamic Mask Attention within modern efficient operator frameworks, we benchmarked the forward, decoding, and backward performance of MHA, SWA, MLA, NSA, and DMA on an A100-SXM4-80GB GPU. The results represent the average of 1,000 runs after three warm-up iterations; specific configurations and implementations can be found in Table 5. As shown in Figure 5, DMA provides robust acceleration across multiple key stages, demonstrating its significant scaling advantages for long contexts.

**Forward**: The time consumption of DMA grows approximately linearly with sequence length (keeping constant $w$), while MHA and MLA exhibit quadratic growth. Compared to MHA, DMA achieves speedups of approximately $10.5\times$, $26.1\times$, $10.2\times$, and $21.5\times$ at token lengths of $4096/8192/16384/32768$, respectively. In the small length range $\leq 2048$, MHA has not yet entered a significant quadratic bandwidth bottleneck, resulting in smaller speedups; however, from 4096 onwards, the quadratic term dominates, and DMA's advantage rapidly amplifies. Compared to SWA, DMA has extremely low overhead for global top-$w$ sampling, with performance converging or slightly better for lengths $\geq 4096$; MLA has a lower constant at small lengths but degrades significantly with length; NSA is noticeably slower than DMA in the medium to long length range due to multi-branch and block scheduling overhead.

**Decode**: In the decoding phase, where complexity degrades to accessing the retained key set, DMA exhibits exponential relative optimization at extremely long contexts: speedups against MHA at key lengths of $8,192/16,384/32,768/65,536/131,072/262,144/524,288$ are approximately $4.0\times$, $6.5\times$, $12.7\times$, $23.3\times$, $49.6\times$, $92.7\times$, and $171.1\times$, respectively. The speedup relative to NSA also continues to expand at the same lengths (e.g., approximately $47.0\times$ at 524,288 keys), indicating that the content-aware constant window significantly reduces throughput and cache pressure during the ultra-long context KV access phase.

**Backward**: The backward phase of DMA also maintains a linear scaling trend, achieving speedups against MHA of approximately $2.5\times$, $4.4\times$, and $7.9\times$ at lengths of $8192/16384/32768$, respectively. Compared to the forward pass, the quadratic cost in the backward pass is exposed earlier due to the need to reuse or recompute attention scores during gradient backpropagation; DMA avoids large-scale redundant score and softmax backpropagation overhead by explicitly skipping masked blocks, thereby maintaining stable bandwidth utilization.

### 3.4 PERFORMANCE COMPARISON

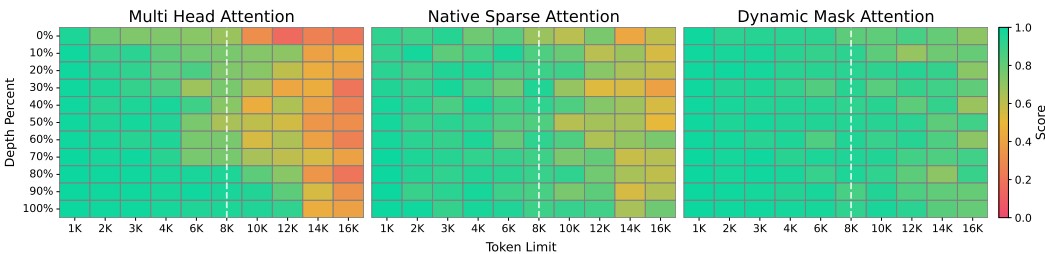

Figure 6: **Needle in a Haystack Performance**. Comparison of MHA, NSA, and DMA in an apples-to-apples setting. The white dotted line indicates the sequence length of the model.

Table 1: **Downstream Task Evaluations for Base Model**. The best results for each size are in bold, and the second-best results are unlined. For the pre-trained base model, DMA outperforms MHA and NSA, as well as other advanced sparse inference methods in most tasks.

| MODEL | PILE PPL ↓ | LAMBADA PPL ↓ | LAMBADA ACC ↑ | MMLU ACC ↑ | TRIVIAQA ACC ↑ | ARC ACC ↑ | PIQA ACC ↑ | HELLASWAG ACC ↑ | OBQA ACC ↑ | WINOGRANDE ACC ↑ | LONGBENCH AVG ↑ |
|---|---|---|---|---|---|---|---|---|---|---|---|
| | | | | | Zero-Shot | | | | | | |
| MHA | 48.65 | 15.22 | 44.3 | 35.4 | 9.4 | 53.4 | 72.9 | 56.1 | 37.0 | 57.3 | 14.2 |
| H2O | — | 15.38 | 44.2 | 34.8 | 7.4 | 53.3 | 72.8 | 55.6 | 36.6 | 56.9 | 8.7 |
| InfLLM | 48.96 | 15.23 | 44.2 | 35.1 | 8.0 | 53.1 | 72.4 | 55.8 | 36.6 | 56.8 | 9.2 |
| Quest | 49.68 | 15.43 | 43.9 | 35.1 | 7.6 | 53.1 | 72.6 | 56.1 | 36.8 | 57.2 | 9.6 |
| DAM | 49.72 | 15.89 | 44.5 | 34.6 | 8.9 | 52.1 | 72.3 | 56.2 | 36.3 | 56.0 | 10.4 |
| Exact-Top | 53.31 | 15.23 | 44.4 | 35.3 | 9.2 | 53.3 | 72.8 | 56.0 | 36.8 | 57.0 | 13.8 |
| NSA | 48.73 | 14.91 | 45.2 | 33.8 | 8.7 | 53.1 | 72.8 | 56.7 | 36.3 | 57.8 | 15.4 |
| DMA (ours) | 45.12 | 14.42 | 45.9 | 37.0 | 9.1 | 55.6 | 73.4 | 56.4 | 36.5 | 58.4 | 16.2 |
| | | | | | Five-Shot | | | | | | |
| MHA | — | 19.40 | 40.4 | 36.8 | 13.2 | 56.8 | 73.2 | 56.8 | 38.0 | 58.6 | — |
| H2O | — | 19.14 | 38.9 | 35.7 | 10.5 | 56.6 | 73.2 | 56.4 | 37.8 | 58.1 | — |
| InfLLM | — | 19.13 | 41.3 | 35.9 | 11.7 | 56.7 | 73.3 | 56.1 | 38.0 | 57.7 | — |
| Quest | — | 19.22 | 40.9 | 36.1 | 10.9 | 56.2 | 73.2 | 55.8 | 37.9 | 58.2 | — |
| DAM | — | 19.47 | 41.2 | 35.2 | 13.3 | 55.1 | 71.0 | 54.4 | 38.0 | 57.2 | — |
| Exact-Top | — | 18.22 | 39.7 | 36.4 | 13.1 | 56.3 | 73.4 | 56.5 | 38.2 | 58.5 | — |
| NSA | — | 21.37 | 39.6 | 34.6 | 12.5 | 56.1 | 76.0 | 58.9 | 39.2 | 58.3 | — |
| DMA (ours) | — | 17.88 | 40.9 | 38.2 | 12.6 | 56.4 | 76.6 | 58.7 | 39.6 | 60.4 | — |

Table 2: **Downstream Evaluations for Finetuned Model**. The best results are in bold, and the second-best results are underlined. For models supervisedly fine-tuned at a 16K sequence length, DMA outperforms other methods in most tasks.

| METHOD | MMLU ACC ↑ | BBH ACC ↑ | GSM8K ACC ↑ | MATH ACC ↑ | MBPP ACC ↑ | LONGBENCH ACC ↑ | RULER ACC ↑ | AVG ACC ↑ |
|---|---|---|---|---|---|---|---|---|
| MHA | 46.4 | 37.7 | 46.3 | 11.7 | 40.0 | 30.2 | 60.6 | 39.0 |
| H2O | 42.4 | 34.5 | 44.8 | 10.5 | 38.5 | 28.5 | 47.8 | 35.3 |
| InfLLM | 44.2 | 36.0 | 45.5 | 11.0 | 39.2 | 29.1 | 55.3 | 37.2 |
| Quest | 44.0 | 34.8 | 45.0 | 10.8 | 38.8 | 28.9 | 50.7 | 36.1 |
| DAM | 45.0 | 36.5 | 46.0 | 11.5 | 37.2 | 26.4 | 49.2 | 36.0 |
| Exact-Top | 45.1 | 37.2 | 45.1 | 11.5 | 38.4 | 28.3 | 44.8 | 35.7 |
| NSA | 43.8 | 38.4 | 46.2 | 11.6 | 40.5 | 30.2 | 59.6 | 38.6 |
| DMA (ours) | 46.2 | 38.2 | 46.8 | 11.6 | 40.6 | 30.7 | 60.5 | 39.2 |

**Downstream Benchmark Evaluations for Base Model.** We adopt the Qwen3 1.7B (Team, 2025) architecture as our base model, modifying only its self-attention component for a fair comparison. The models underwent a two-stage pre-training paradigm: an initial phase on 32 billion tokens with a 2K sequence length to build foundational knowledge, followed by a second phase on 8 billion tokens with an 8K sequence length and an adjusted RoPE base frequency to enhance long-context capabilities (Xiong et al., 2023). This process yielded our three core models: MHA (Vaswani et al., 2017), NSA (Yuan et al., 2025), and DMA. For a comprehensive evaluation, we also compared them against several advanced sparse attention methods based on KV selection, including H2O (Zhang et al., 2023), infLLM (Xiao et al., 2024a), Quest (Tang et al., 2024), DAM (Zhang et al., 2025), and Exact-Top. All these models were evaluated on a wide range of benchmarks, including Pile (Gao et al., 2020), LLAMBADA (Paperno et al., 2016), MMLU (Hendrycks et al., 2021), TriviaQA (Joshi et al., 2017), ARC (Clark et al., 2018), PIQA (Bisk et al., 2020), HellaSwag (Zellers et al., 2019), OBQA (Mihaylov et al., 2018), Winogrande (Sakaguchi et al., 2021), and LongBench (Bai et al.,

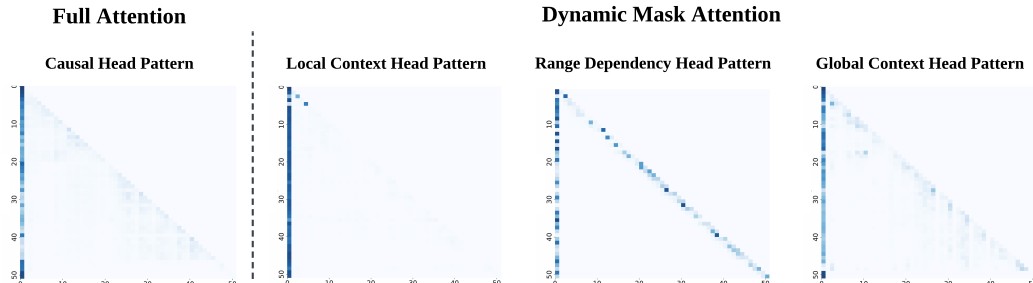

Figure 7: **Dynamic Mask Attention Heatmaps**. The heatmaps show the attention weights of each head in the Dynamic Mask Attention mechanism, indicating which tokens each head focuses on.

2023). The results, shown in Table 1, demonstrate that DMA outperforms the baseline on most tasks in both zero-shot and five-shot settings.

**Downstream Benchmark Evaluations for Finetuned Model.** We further fine-tuned all models at a 16K sequence length with an adjusted RoPE base frequency to 400K, enhancing their long-context generalization capabilities. The models were evaluated on MMLU (Hendrycks et al., 2021), BBH (Suzgun et al., 2023), GSM8K (Cobbe et al., 2021), MATH (Hendrycks et al., 2020), MBPP (Austin et al., 2021), LongBench (Bai et al., 2023), and RULER (Hsieh et al., 2024). As shown in Table 2, DMA achieved the best average score, leading in GSM8K, MBPP, and Long-Bench, while remaining highly competitive in MMLU and RULER (second-best). NSA ranked first in BBH, with DMA closely following, while full-attention MHA performed best on RULER but lagged behind DMA in average score. These results indicate that DMA's content-aware sparse mask effectively transfers even under longer-context fine-tuning.

**Extrapolated Content Retrieval.** We conducted an apples-to-apples comparison of MHA, NSA, and DMA on the needle-in-a-haystack task (Kamradt, 2023) to evaluate their long-context information retrieval capabilities. As shown in Figure 6, DMA's advantage over the other models grows as the context length increases. Crucially, when the context length exceeds the model's pre-training sequence length, the performance of both MHA and NSA degrades significantly, whereas DMA's performance decline is much smaller. This demonstrates DMA's superior length extrapolation capabilities, enabling it to effectively retrieve information from documents that are longer than those seen during training. We attribute this advantage to its content-aware dynamic mask mechanism, which is highly valuable for real-world applications that require precise information extraction from vast texts.

## 4 ANALYSIS

In the Section G, we further analyze the head specialization behavior of Dynamic Mask Attention. We visualize the attention heatmaps 7 and show that different heads naturally develop diverse sparse patterns, including local context heads focusing on nearby tokens, range dependency heads capturing long-range semantic links, and global context heads aggregating information across the entire sequence. This content-aware specialization allows DMA to adapt its attention strategy dynamically to different inputs, providing stronger long-range modeling and more efficient multi-scale context integration compared to traditional dense attention.

## 5 CONCLUSION

We presented Dynamic Mask Attention (DMA), a trainable content-aware sparse attention mechanism for long-context modeling. DMA combines content-driven mask generation with Sparse Weights Computation, enabling adaptivity and end-to-end differentiability. Soft-gating masks and a differentiable mask generator overcome the hard binary and non-trainable limitations of prior sparse methods. Extensive experiments demonstrate that DMA achieves consistent improvements across model scales and evaluation settings. It reduces perplexity compared to strong baselines, improves associative recall, and substantially accelerates both training and inference, with up to $50\times$ training speedup and $10\times$ inference speedup on long sequences.

## ETHICS STATEMENT

We have read and will adhere to the ICLR Code of Ethics and the ICLR Code of Conduct. Our research introduces ReCode, a framework for LLM-powered agents, and evaluates it within simulated environments. The datasets used in our study are well-established public benchmarks for academic research. These environments do not contain any personally identifiable information (PII) or sensitive real-world data. Our work did not involve human subjects, crowd-sourcing, or the scraping of private data; therefore, Institutional Review Board (IRB) approval was not required.

We acknowledge that research on autonomous agents carries potential dual-use risks. To mitigate these, our experiments are intentionally confined to benign, closed-world tasks such as online shopping and household activities within simulated settings. We followed good scholarly practice by reporting our methods and results transparently and citing prior work accurately. The authors declare no competing interests or external sponsorships that could have influenced the outcomes of this research.

## REPRODUCIBILITY STATEMENT

We are committed to ensuring the reproducibility of our research. All essential details for reproducing our results are provided within this paper. The descriptions of the datasets (ALFWorld, WebShop, ScienceWorld) and their respective splits are detailed in the Appendix. Our experimental setup, including the specific models used for inference and fine-tuning, training configurations, and evaluation protocols, is described in Section F. To facilitate full replication, we will release our code and few-shot prompts as supplementary material. An anonymous repository containing these artifacts is available at: `https://github.com/dma-anonymity/dma`.

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

APPENDIX

# A  THE USE OF LARGE LANGUAGE MODELS (LLMS)

We used Large Language Models (LLMs), specifically Gemini 2.5 Pro, Claude Sonnet 4, and GPT-5, solely as assistive tools for grammar correction and minor stylistic edits to improve the manuscript's clarity and logical flow. The LLMs did not generate, modify, or determine any scientific ideas, methods, experiments, analyses, results, or conclusions. All technical content was written and verified by the authors.

To preserve anonymity and confidentiality, no identifying information or nonpublic materials were shared with any LLM service. Text provided for editing was de-identified. All LLM suggestions were reviewed by at least one author before incorporation, and any unverifiable suggestions were discarded. The authors take full responsibility for the content of this paper.

# B  RETHINKING THE SPARSE ATTENTION MECHANISMS

As the context windows of Large Language Models (LLMs) expand from thousands to millions of tokens, the quadratic complexity of self-attention has emerged as a formidable computational hurdle. To surmount this challenge, sparse attention has become a critical optimization. A dominant paradigm within this domain relies on content-agnostic methods, which employ fixed sparsity patterns such as sliding windows and strided attention. While appealing for their structural simplicity and ease of implementation, their fundamental deficiency lies in ignoring input semantics; they operate on the rigid assumption that relevance is a function of positional proximity. This inherent limitation has paved the way for an alternative design philosophy: content-aware sparse attention.

Instead of adhering to static, predetermined patterns, this paradigm dynamically constructs attention pathways based on the semantic substance of the input itself. This allows for a far more robust and flexible capture of long-range dependencies. Consequently, a comparative analysis of content-agnostic versus content-aware is fundamental to understanding the evolution of sparse attention mechanisms.

## B.1  CONTENT-AGNOSTIC SPARSITY ATTENTION

Content-agnostic sparse attention methods rely on a predefined, fixed sparsity pattern and apply it uniformly to all inputs regardless of their semantic content. These methods, such as Sliding Window Attention (SWA) (Fu et al., 2025), Strided Attention Child et al. (2019), and BigBird Zaheer et al. (2020), operate under the simplifying assumption that token relevance is primarily determined by positional proximity. SWA restricts each token to attending only to neighboring tokens within a local window. While this effectively captures short-range dependencies, it cannot handle long-range semantic connections beyond the window boundaries. Strided Attention selects tokens at fixed intervals, shortening the sequence length with a fixed stride. While computationally efficient, it can miss key tokens that do not conform to the stride pattern, especially in documents with irregular distributions of key information. BigBird attempts to balance context at different scales by combining local, global, and random attention into a hybrid structure. However, its weights and patterns remain predetermined and static, lacking adaptability to the specific semantics of the input.

## B.2  CONTENT-AWARE SPARSITY ATTENTION

To overcome the limitations of fixed sparsity patterns, content-aware sparse attention has been proposed. Its core goal is to drive sparsity based on the semantic importance of content, rather than simple structured rules. Existing content-aware methods can be roughly divided into two categories.

**Heuristic Rule-Based Methods.**  These methods dynamically select important tokens using preset rules. For example, H2O (Zhang et al., 2023) retains high-weight tokens based on the "heavy hitter" assumption, StreamingLLM (Xiao et al., 2024b) combines an attention sink with a local window to dynamically select tokens, and SnapKV (Li et al., 2024) filters tokens based on importance scores. FlexPrefill (Lai et al., 2025) further introduces a query-aware mechanism using Jensen-Shannon

divergence to switch between patterns. While these methods achieve content-aware adaptation, their primary limitation is that the selection rules rely on fixed heuristics or statistical metrics that cannot be optimized via gradient descent.

**Staged Computation methods.** This kind of method adopts a two-stage design: First, they compute an importance metric and then generate a sparse mask based on this metric. For example, DAM (Zhang et al., 2025) generates dynamic masks by analyzing attention heatmaps, while Quest (Tang et al., 2024) first evaluates the importance of token blocks and then selects the top-k blocks for computation. Similarly, MoBA (Lu et al., 2025) employs a routing mechanism to select attention blocks, akin to Mixture-of-Experts. The main drawback of these approaches is the decoupled nature of mask generation or coarse-grained block selection, which leads to increased computational overhead and inconsistent optimization objectives.

These distinct limitations highlight a shared, fundamental problem: *the inability to optimize the sparse attention pattern in a fully end-to-end manner.* Native Sparse Attention (Yuan et al., 2025) makes significant progress on this front by introducing a trainable sparse attention mechanism. However, it still falls short of being end-to-end optimizable. This is because NSA's architecture is composed of three distinct branches, which still incorporate fixed, content-agnostic sparse patterns like local or strided attention. Consequently, only a part of the overall mechanism is truly optimizable, preventing a fully adaptive optimization based on content. Therefore, the critical challenge remains: to develop a sparse attention that is not just partially trainable, but fully differentiable and learns its structure from scratch based on semantic content.

## C BACKGROUND

### C.1 LANGUAGE MODELING TASKS

**Sparsity in Language Modeling.** As shown in Figure 8, long-context language modeling involves three fundamental tasks: Copying, Selecting, and Inducing. The Copy task requires the model to maintain a fixed-distance relationship between the input and output; the Select task requires the model to remember or ignore specific elements based on their content selectively; and the Induce task requires the model to retrieve answers through associative recall based on context. These three tasks naturally exhibit different sparsity patterns: positional sparsity in the Copy task, where only tokens at fixed distances are attended to; content sparsity in the Select task, where only tokens with specific content are attended to; and associative sparsity in the Induce task, where only key-value pairs relevant to the query are attended to. These inherent sparsity patterns provide a theoretical foundation for designing more efficient attention mechanisms.

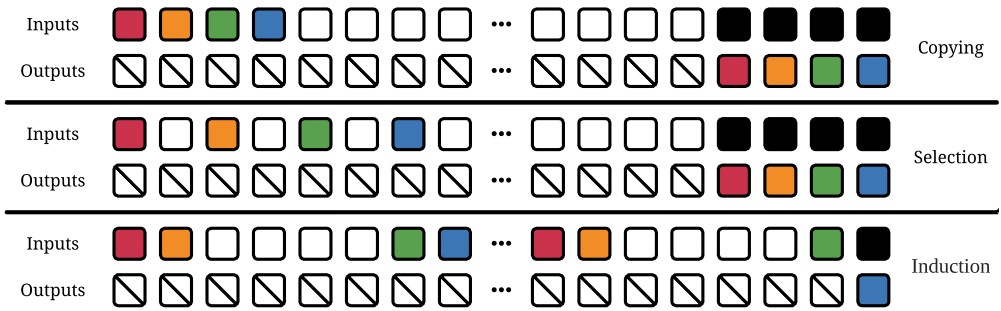

Figure 8: **Sparsity in Language Modeling Tasks**. The tasks of Copy (Romero et al., 2021), Select (Arjovsky et al., 2016), and Induce (Olsson et al., 2022) are three essential tasks for language modeling. The Copy task requires maintaining a fixed distance between input elements and output elements, the Select task involves selectively remembering or ignoring certain elements based on the input, and the Induce task requires retrieving answers through associative recall based on context. Where the colored parts represent the tokens that the model needs to remember in the current time step of inference, the black parts represent the output tokens that the model needs to predict based on the input, and the white parts represent irrelevant tokens that can be filtered out.

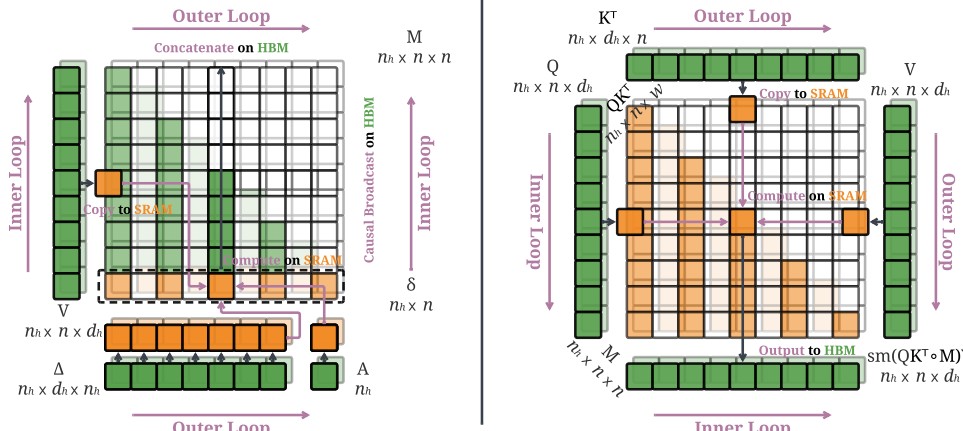

Figure 9: **Dynamic Mask Attention Architecture**. **Left**: **Step 1: Dynamic Mask Generation.** The mask computation part of the dynamic mask attention. **Right**: **Step 2: Sparse Weights Computation.** The weight computation part of the dynamic mask attention.

### C.2 MULTI-HEAD ATTENTION

**QKV Projection.** In the Transformer, we first convert the input into query, key, and value representations. For the hidden state $h_t \in \mathbb{R}^{d_{model}}$ of the $t$-th token, we perform linear projections through the weight matrices $W^Q$, $W^K$, and $W^V$ to obtain $q_t$, $k_t$, and $v_t$ respectively, as shown in Equation 19. These projections transform the input representation into separate subspaces for each attention head, allowing each head to focus on distinct aspects of the input. The weight matrices shape the projection into $n_h$ attention heads, each with a dimension of $d_h$, where $d_{model} = n_h \times d_h$ is typically held.

$$
\begin{aligned}
q_t &= h_t W^Q \quad where \quad h_t \in \mathbb{R}^{d_{model}} \quad W^Q \in \mathbb{R}^{d_{model} \times n_h \times d_h} \quad q_t \in \mathbb{R}^{n_h \times d_h} \\
k_t &= h_t W^K \quad where \quad h_t \in \mathbb{R}^{d_{model}} \quad W^K \in \mathbb{R}^{d_{model} \times n_h \times d_h} \quad k_t \in \mathbb{R}^{n_h \times d_h} \\
v_t &= h_t W^V \quad where \quad h_t \in \mathbb{R}^{d_{model}} \quad W^V \in \mathbb{R}^{d_{model} \times n_h \times d_h} \quad v_t \in \mathbb{R}^{n_h \times d_h}
\end{aligned}
\tag{19}
$$

**Key-Value Concatenation.** In autoregressive generation, we need to cache the key-value pairs of historical tokens to avoid redundant computations. As shown in Equation 20, we concatenate the cached past token key matrix and value matrix with the current token's key-value representations to form the complete key matrix $k$ and value matrix $v$. By maintaining and updating this cache, we construct a complete context window that spans all tokens from position 1 to the current position $t$, enabling the model to access and utilize the full sequence history.

$$
\begin{aligned}
k &= \text{concat}([k_1, \ldots, k_t]) \quad where \quad k \in \mathbb{R}^{n_h \times t \times d_h} \\
v &= \text{concat}([v_1, \ldots, v_t]) \quad where \quad v \in \mathbb{R}^{n_h \times t \times d_h}
\end{aligned}
\tag{20}
$$

## D PROOF OF METHODOLOGY

We use $h_t$ to represent the $t$-th token, $n$ to represent the sequence length, $d$ to represent the embedding dimension, $n_h$ to represent the number of heads, and $[., \ldots, .]$ and $[., \vdots, .]$ to represent concatenation. Assuming that we have completed the linear projection of $h_t$ for the query matrix, key matrix, and value matrix, denoted as $q_t, k_t, v_t \in \mathbb{R}^{n_h \times d_h}$, and concatenated the kv cache to obtain $k, v \in \mathbb{R}^{n_h \times n \times d_h}$, we derive the computation pipeline for the $t$-th token.

### D.1 DYNAMIC MASK GENERATION

We first need to generate a set of content-aware dynamic masks for each attention head at the current time step, which will guide the subsequent attention weight computation to focus only on the most important key positions.

$$\delta = \exp(\tau(v \cdot \Delta) \times A) \tag{21}$$

We introduce the sampling stride weight $\Delta \in \mathbb{R}^{n_h \times d_h \times n_h}$ and head-by-head gate coefficient $A \in \mathbb{R}^{n_h}$, as well as a non-negative range activation function $\tau(\cdot)$, to sample the value vector representations. As shown in equation 21, we first perform a tensor contraction $v \cdot \Delta \in \mathbb{R}^{n_h \times n}$, converting each token's $d_h$-dimensional content vector into a scalar feature as a preliminary estimate of its importance. Subsequently, the non-negative activation function $\tau(\cdot)$ ensures that the scores are non-negative, avoiding signal suppression. Then, the gate coefficient $A$ for each head scales the importance scores, allowing the model to learn different sparsity levels for different heads. Finally, the exponential function $exp(\cdot)$ maps the final scores to the positive value space and amplifies the differences between high and low scores, facilitating the learning of gating effects. The final scores $\delta \in \mathbb{R}^{n_h \times n}$ are obtained.

$$
\begin{aligned}
m_t &= f(\delta) \\
&= \begin{bmatrix} f(\sum_{j=1}^{t} \delta_{1,j}) \\ f(\sum_{j=1}^{t} \delta_{2,j}) \\ \vdots \\ f(\sum_{j=1}^{t} \delta_{n_h,j}) \end{bmatrix} \quad where \quad f(\delta_{n_h,j}) = \begin{cases} \delta_{n_h,j} & \text{if } \delta_{n_h,j} \in top(\delta_{n_h}, w) \\ -\infty & \text{otherwise} \end{cases}
\end{aligned} \tag{22}
$$

Subsequently, as shown in equation 22, we define the operation set $f(\cdot)$ to determine whether each head's score $\delta_{n_h,j}$ is in the top-$w$ of that head. If it is, its original score is retained to ensure that gradients can be computed; otherwise, it is set to $-\infty$ to ensure that its probability is 0 in the subsequent softmax. If it is causal language modeling, we can also introduce causal broadcasting in the operation set to ensure no additional memory overhead. The final mask $m_t \in \mathbb{R}^{n_h \times n}$ is obtained.

This approach has three obvious advantages. First, sampling importance scores from value representations can more accurately focus on semantically critical tokens that are distant but important, alleviating the problem of getting lost compared to pure positional patterns. Second, through $A$ and independent $top_w$, different heads can spontaneously differentiate into diverse functions such as local, remote, and global, improving representation coverage. Last, sparse testing is natively effective during the training phase, without the need for posterior pruning, avoiding the disruption of already learned retrieval capabilities. In kernel design, the entire computation flow is shown in the left of Figure 9, in the outer loop, the stride weight $\Delta$ and gate weight $A$ are loaded into high-speed SRAM, and in the inner loop, the zero-order hold method is used to loop through the $V$ blocks loaded into SRAM, sampling from it to generate content-aware $K$ masks. These masks are then causally broadcast to the length of $Q$ in High Bandwidth Memory (HBM) to avoid quadratic complexity memory usage. Finally, in the outer loop, all mask blocks are concatenated to form the final content-aware sparse dynamic mask.

### D.2 SPARSE WEIGHTS COMPUTATION

After obtaining the dynamic mask, we use it to sparsify the scaled dot-product weight computation process of attention, reducing the single-step complexity from $O(nd_h)$ to $O(wd_h)$.

$$o_t = \text{softmax}(q_t \cdot k^\top \circ m_t) \cdot v$$

$$= \begin{bmatrix} \sum_{j=1}^{t} p_{1,j} \cdot v_{1,j} \\ \sum_{j=1}^{t} p_{2,j} \cdot v_{2,j} \\ \vdots \\ \sum_{j=1}^{t} p_{n_h,j} \cdot v_{n_h,j} \end{bmatrix} \quad where \quad p_{n_h,j} = \begin{cases} \frac{\exp(q_{n_h} \cdot k_{n_h,j}^\top + m_{n_h,j})}{\sum_{j'=1}^{t} \exp(q_{n_h} \cdot k_{n_h,j'}^\top + m_{n_h,j'})} & \text{if } m_{n_h,j} \neq -\infty \\ 0 & \text{if } m_{n_h,j} = -\infty \end{cases}$$

$$(23)$$

For the $t$-th query $q_t \in \mathbb{R}^{n_h \times d_h}$ and all keys and values $k, v \in \mathbb{R}^{n_h \times n \times d_h}$, the entire computation flow is shown in equation 23. Since the scores at positions marked as $-\infty$ by $m_t$ have probabilities close to 0 after $softmax$, the key vectors at these positions will not affect the final output, so we can skip all computations at the $(h, j)$ position. For the retained positions, we compute their scaled dot-product attention scores $q_t k^\top \in \mathbb{R}^{n_h \times n}$ and add the sampled scores retained in $m_t$ to these scores. The above operation allows the multi-head mechanism to work with the dynamic mask, enabling the model to focus on different patterns in the sequence in parallel. These context vectors are concatenated to produce the final attention output $o_t \in \mathbb{R}^{n_h \times d_h}$. This output integrates information from all attention heads, forming a multi-level contextual representation that effectively captures dependencies at different distances in the sequence history. It can approximate a complete attention mechanism under the condition of $n_h \times w < n$, while maintaining computational efficiency.

This approach also has three significant advantages. First, the mask structurally reduces the candidate set participating in matrix multiplication and softmax in advance, avoiding pseudo-sparsity that zeros out after full computation. Second, unlike KV selection methods, we do not delete the original sequence's $k, v$, so other heads can still access the global context when needed. Finally, in kernel implementation, block masks can be directly loaded for judgment; if all positions in the current block are masked, block-level skipping can be executed directly, avoiding shared memory loading and matrix multiplication accumulation computations. In kernel design, the entire computation flow is shown in the right of Figure 9, where in the outer loop, the $K$ and $V$ blocks are looped and loaded into SRAM, and in the inner loop, the $Q$ blocks are accessed, loaded into SRAM, and the output of the attention weight computation is written back to HBM. If the current position of the $K$ block is designated as masked in the dynamic mask, the attention weight at that position is directly filled with 0, skipping the computation at that position, forming the final position-aware sparse attention weights.

### D.3 FULLY GRADIENT FLOW

Finally, we ensure that the introduced dynamic mask and sparse weights do not block gradients, and the gradients of the retained attention paths are strictly consistent with those of full attention. They can flow completely to all inputs and parameters without gradient discontinuity issues caused by discrete operations, supporting end-to-end training and aligning with our goal of preserving key dependencies while suppressing redundant costs.

For clarity, we derive for a single time step $t$ and a single attention head $h$, with multiple heads being parallel along the $h$ dimension. Let the set of selected indices be $\mathcal{I}_h \in \mathbb{Z}^w$, and unselected positions are treated as adding $m_{h,j} = -\infty$, so that $p_{h,j} \approx 0$ and do not participate in any multiplication. We denote the intermediate quantities in the forward pass for subsequent gradient derivation:

$$s_{h,j} = \frac{q_h \cdot k_{h,j}}{\sqrt{d_h}} + m_{h,j}, \quad p_{h,j} = \begin{cases} \frac{\exp(s_{h,j})}{\sum\limits_{j' \in \mathcal{I}_h} \exp(s_{h,j'})}, & j \in \mathcal{I}_h \\ 0, & j \notin \mathcal{I}_h \end{cases}, \quad o_h = \sum_{j \in \mathcal{I}_h} p_{h,j} v_{h,j}. \quad (24)$$

Let the upstream gradient of the loss $L$ arriving at the head output be $g_h = \frac{\partial L}{\partial o_h} \in \mathbb{R}^{d_h}$. Then the gradient of $v$ is that $g_h$ is distributed to the selected $v_{h,j}$ according to the proportion of $p_{h,j}$, and the gradient at unselected positions is 0.

$$\frac{\partial L}{\partial v_{h,j}} = p_{h,j}\, g_h \quad (j \in \mathcal{I}_h), \qquad \frac{\partial L}{\partial v_{h,j}} = 0 \ (j \notin \mathcal{I}_h) \tag{25}$$

Let $dp_{h,j} = \frac{\partial L}{\partial p_{h,j}}$. The standard softmax Jacobian is $\frac{\partial p_{h,k}}{\partial s_{h,j}} = p_{h,k}(\delta_{jk} - p_{h,j})$. Substituting into the chain rule gives $\frac{\partial L}{\partial s_{h,j}} = \sum_k dp_{h,k} p_{h,k}(\delta_{jk} - p_{h,j}) = p_{h,j}\big(dp_{h,j} - \sum_k p_{h,k} dp_{h,k}\big)$. Letting $\alpha_h = \sum_k p_{h,k} dp_{h,k}$, the element-wise form is $ds_{h,j} = p_{h,j}(dp_{h,j} - \alpha_h)$. Masked positions naturally get $ds_{h,j} = 0$ because $p_{h,j} = 0$, without explicit computation.

$$ds_{h,j} = (dp_{h,j} - \sum_{j' \in \mathcal{I}_h} p_{h,j'} \circ dp_{h,j'}) \times p_{h,j} \tag{26}$$

Because $s_{h,j} = q_h \cdot k_{h,j} + m_{h,j}$, by additive decomposition

$$\frac{\partial L}{\partial m_{h,j}} = ds_{h,j}. \tag{27}$$

If $j \notin \mathcal{I}_h$, then $ds_{h,j} = 0$, and the mask gradient is naturally 0, without needing to differentiate the top-$w$ decision, treating it as a constant graph structure selected in the forward pass.

For the gradients of $q, k$, since $s_{h,j}$ linearly depends on $q_h, k_{h,j}$, substituting into the chain rule gives

$$\frac{\partial L}{\partial q_h} = \sum_{j \in \mathcal{I}_h} ds_{h,j} \frac{k_{h,j}}{\sqrt{d_h}}, \qquad \frac{\partial L}{\partial k_{h,j}} = ds_{h,j} \frac{q_h}{\sqrt{d_h}} \tag{28}$$

Our approach has several significant advantages. First, for the selected positions, the gradients are identical to those of full attention, and DMA only prunes the operator chain for positions whose contributions can be ignored, ensuring expressiveness. Then, only second-order correlation information is propagated to $\mathcal{I}_h$, improving bandwidth utilization. The gating parameter $A$ and weight $\Delta$ directly receive semantic feedback from attention scores, quickly shaping local, remote, and global specialization. Finally, the equivalence relation $dM = dS$ allows the kernel to only recompute the local $S$ without storing additional intermediate mask gradient tensors, improving composability.

## E  IMPLEMENTATION DETAILS OF DYNAMIC MASK ATTENTION

The following listing provides a sample implementation of the Dynamic Mask Attention algorithm in PyTorch, as described in Section 2.

The implementation demonstrates the core computational flow of the Dynamic Mask Attention mechanism. First, the query, key, and value matrices are computed through linear projections, followed by the application of rotary position embeddings. The core innovation of the algorithm is then reflected in the dynamic mask generation process: dynamic weights $\delta$ are calculated from the value vectors, and a sparse mask is generated using the topk operation, retaining only the most relevant $w$ key-value pairs. Finally, in the sparse attention computation phase, the algorithm computes attention weights only for the selected key-value pairs, significantly reducing computational complexity. In actual kernel implementations, it is possible to check if there are any active tokens in the MMA block; if not, the computation for that block can be skipped.

**Listing 1** The Implementation of Dynamic Mask Attention in PyTorch

```python
def dynamic_mask_attention(h_t, position_embed, causal_m,
↪    kv_cache,
    W_Q, W_K, W_V, W_dt, A, W_O, W):
    # linear projections
    q_t = W_Q(h_t)...                    # [b, n_h, q_len, d_h]
    k_t = W_K(h_t)...                    # [b, n_h, q_len, d_h]
    v_t = W_V(h_t)...                    # [b, n_h, q_len, d_h]
    o_t = torch.zeros_like(q_t)          # [b, n_h, q_len, d_h]
    # apply rotary position embeddings
    q_t, k_t = apply_rope(q_t, k_t, *position_embed)
    # concatenate past key and value states
    k, v = kv_cache.update(k_t, v_t)     # [b, n_h, k_len, d_h]
    # calculate dynamic mask
    dt = W_dt(v...)                      # [b, k_len, n_h]
    dt = torch.exp(A * F.softplus(dt))... # [b, n_h, k_len]
    m_t = dt.expand(-1, -1, q_len, -1)   # [b, n_h, q_len, k_len]
    active_m = torch.zeros_like(m_t)
    m_t = m_t.masked_fill(causal_m != 0, -float('inf'))
    topk_indices = torch.topk(m_t, W).indices
    active_m = active_m.scatter(-1, topk_indices, 1.0)
    m_t = m_t.masked_fill(active_m == 0.0, -float('inf'))
    # calculate sparse attention weight
    for b_idx in range(b):
        for h_idx in range(n_h):
            for q_idx in range(q_len):
                q_elem = q_t[b_idx,h_idx,q_idx,:]        # [d_h]
                w_idx = topk_indices[b_idx,h_idx,q_idx]  # [w]
                k_vecs = k[b_idx,h_idx,w_idx,:]          # [w, d_h]
                v_vecs = v[b_idx,h_idx,w_idx,:]          # [w, d_h]
                m_elems = m_t[b_idx,h_idx,q_idx,w_idx]   # [w]
                a_elem = sum(q_elem... * k_vecs)         # [w]
                a_elem = a_elem / sqrt(d_h) + m_elems
                a_elem = softmax(a_elem)
                o_elem = sum(a_elem... * v_vecs)         # [d_h]
                o_t[b_idx,h_idx,q_idx,:] = o_elem
    o_t = o_t...          # [b, q_len, n_h, d_h]
    h_t = W_O(o_t...)     # [b, q_len, d_model]
    return h_t
```

## F  EXPERIMENT SETTINGS

To make the comparison between attention variants fair and reproducible, we standardize the model, data pipeline, optimization, and evaluation, only changing the attention module and its related hyperparameters. All experiments were conducted using the open-source PyTorch images (NVIDIA, 2022) and the Transformers framework (Wolf et al., 2020). We use SmolLMCorpus (Ben Allal et al., 2024) as training data. For evaluation frameworks, we utilized the LM evaluation harness (Gao et al., 2021) from EleutherAI for perplexity tasks, and the lighteval (Fourrier et al., 2023) from Hugging-Face for downstream tasks. Table 3 lists the model sizes and key hyperparameters used at each scale. We keep the depth, width, and number of heads consistent across variants under the same parameter budget, only changing the specific parameters of the attention variants, with parameter symbols consistent with those in the original papers of the attention variants.

We summarize the meaning of the columns in Table 3 and clarify which hyperparameters are used by each attention variant.

• **Params**: total number of model parameters.

Table 3: **Self-Attention Variants Scaling Laws Configurations**. The model and hyperparameter configurations used in our self-attention variants scaling laws experiments.

| ALGOS | PARAMS | STEPS | BATCH | LR | $n_{layers}$ | $d_{model}$ | $n_h$ | $n_{h_{kv}}$ | $w$ | $d_c$ | $B$ | $B'$ | $k$ |
|---|---|---|---|---|---|---|---|---|---|---|---|---|---|
| MHA | $\approx 80$M | 13,500 | 0.128M tokens | 3e-3 | 12 | 768 | 6 | 3 | - | - | - | - | - |
| SWA | $\approx 80$M | 13,500 | 0.128M tokens | 3e-3 | 12 | 768 | 6 | 3 | 1024 | - | - | - | - |
| MLA | $\approx 80$M | 13,500 | 0.128M tokens | 3e-3 | 12 | 768 | 6 | 3 | - | 192 | - | - | - |
| NSA | $\approx 80$M | 13,500 | 0.128M tokens | 3e-3 | 12 | 768 | 6 | 3 | 512 | 192 | 32 | 64 | 16 |
| DMA | $\approx 80$M | 13,500 | 0.128M tokens | 3e-3 | 12 | 768 | 6 | 3 | 1024 | - | - | - | - |
| MHA | $\approx 200$M | 20,800 | 0.192M tokens | 2e-3 | 16 | 1024 | 8 | 4 | - | - | - | - | - |
| SWA | $\approx 200$M | 20,800 | 0.192M tokens | 2e-3 | 16 | 1024 | 8 | 4 | 1024 | - | - | - | - |
| MLA | $\approx 200$M | 20,800 | 0.192M tokens | 2e-3 | 16 | 1024 | 8 | 4 | - | 256 | - | - | - |
| NSA | $\approx 200$M | 20,800 | 0.192M tokens | 2e-3 | 16 | 1024 | 8 | 4 | 512 | 192 | 32 | 64 | 16 |
| DMA | $\approx 200$M | 20,800 | 0.192M tokens | 2e-3 | 16 | 1024 | 8 | 4 | 1024 | - | - | - | - |
| MHA | $\approx 680$M | 35,000 | 0.392M tokens | 1e-3 | 24 | 1536 | 12 | 6 | - | - | - | - | - |
| NSA | $\approx 680$M | 35,000 | 0.392M tokens | 1e-3 | 24 | 1536 | 12 | 6 | 512 | 192 | 32 | 64 | 16 |
| DMA | $\approx 680$M | 35,000 | 0.392M tokens | 1e-3 | 24 | 1536 | 12 | 6 | 1024 | - | - | - | - |
| MHA | $\approx 1.7$B | 40,000 | 1M tokens | 1e-3 | 28 | 2048 | 16 | 8 | - | - | - | - | - |
| NSA | $\approx 1.7$B | 40,000 | 1M tokens | 1e-3 | 28 | 2048 | 16 | 8 | 512 | 256 | 32 | 64 | 16 |
| DMA | $\approx 1.7$B | 40,000 | 1M tokens | 1e-3 | 28 | 2048 | 16 | 8 | 2048 | - | - | - | - |

- **Steps**: training steps.

- **Batch**: tokens per step.

- **LR**: peak learning rate.

- $n_{layers}$: number of Transformer layers.

- $d_{model}$: model hidden size.

- $n_h$: number of attention heads.

- $n_{h_{kv}}$: number of KV heads. In all our configurations we set $n_{h_{kv}} = n_h/2$.

- **MHA** (full attention): standard scaled dot-product attention. Sparse-specific columns ($w$, $d_c$, $B$, $B'$, $k$) are not used.

- **SWA** (sliding-window attention): $w$ is the sliding window size; other sparse-specific columns are not used.

- **MLA** (multi-head latent attention): $d_c$ is the latent/compression dimension; other columns are not used.

- **NSA** (native sparse attention): $d_c$ is the compression dimension, $w$ is the sliding window size, $B$ is the compressing block size, $B'$ is the selection block size, and $k$ is the num selected blocks. All settings following the original paper.

- **DMA** (dynamic mask attention): $w$ is the per-head top-$w$ kept keys for dynamic masks; other sparse-specific columns are not used.

For ease of interpretation of Table 4, we compare the per-head theoretical complexities of different attention mechanisms under a unified assumption: default causal attention, fixed head dimension $d_h$, ignoring constant overheads such as QKV/output projection and normalization, and only counting attention weight computation and corresponding activation peak memory. Let $n$ be the sequence length, $w$ be the number of keys retained per head, $d_c$ be the compression/latent variable dimension, $B$ be the block size, and $k$ be the number of selected blocks (or entries). Thus, we have: MHA is $O(n^2 d_h)/O(n^2)$; SWA is $O(nw d_h)/O(nw)$; MLA uses $d_c$ instead of $d_h$ for global computation, with complexity $O(n^2 d_c)/O(n^2)$; NSA's composite terms come from the compression branch $O(n^2 d_c/B)$, selection branch $O(nkB d_h)$, and local branch $O(nw d_h)$, with memory $O(n^2/B + nkB)$; KV selection methods (H2O/InfLLM/Quest/DAM) only compute for the selected $k$ items, yielding $O(nk d_h)/O(nk)$; DMA is in the same order as SWA $O(nw d_h)/O(nw)$, but $w$ comes from content-driven per-head dynamic masks (top-$w$), rather than fixed geometric windows. We use this method to estimate the sparsity ratio of different attention methods to ensure fair comparisons under similar computational budgets.

Table 4: **Comparison of Different Attention Variants**. Comparison of different Self-Attention mechanisms. $n$ denotes sequence length, $d_h$ represents head dimension, $w$ is window size, $d_c$ is compressed dimension, $B$ is compression block size, and $k$ is selection budget. Complexities focus on attention weight computation and memory requirements.

| MECHANISM | COMPUTATION COMPLEXITY | MEMORY COMPLEXITY |
|---|---|---|
| MHA | $O(n^2 d_h)$ | $O(n^2)$ |
| SWA | $O(nwd_h)$ | $O(nw)$ |
| MLA | $O(n^2 d_c)$ | $O(n^2)$ |
| NSA | $O(n^2 d_c/B + nkBd_h + nwd_h)$ | $O(n^2/B + nkB)$ |
| H2O | $O(nkd_h)$ | $O(nk)$ |
| InfLLM | $O(nkd_h)$ | $O(nk)$ |
| Quest | $O(nkd_h)$ | $O(nk)$ |
| DAM | $O(nkd_h)$ | $O(nk)$ |
| **DMA** | $O(nwd_h)$ | $O(nw)$ |

Table 5: **Speed Benchmark Configurations**. The common settings for running time curves and the sparse hyperparameters for each method.

| ALGO | warmups | runs | $n_h$ | $n_{h_{kv}}$ | $d_h$ | $w$ | $d_c$ | $B$ | $B'$ | $k$ | precision |
|---|---|---|---|---|---|---|---|---|---|---|---|
| MHA[1] | 3 | 1,000 | 32 | 8 | 128 | - | - | - | - | - | bf16 |
| SWA[2] | 3 | 1,000 | 32 | 8 | 128 | 1024 | - | - | - | - | bf16 |
| MLA[3] | 3 | 1,000 | 32 | 8 | 128 | - | 192 | - | - | - | bf16 |
| NSA[4] | 3 | 1,000 | 32 | 8 | 128 | 512 | 256 | 32 | 64 | 16 | bf16 |
| DMA | 3 | 1,000 | 32 | 8 | 128 | 1024 | - | - | - | - | bf16 |

Table 5 lists the configurations we used in the speed benchmark tests. We use representative configurations, including the number of heads, head dimensions, and specific hyperparameters for each attention variant, to ensure comparisons are made under fair and reproducible conditions. All experiments were conducted on NVIDIA A100 GPUs, using bf16 precision for testing. The runtime only includes the corresponding computation of attention weights, excluding the time overhead of projections and other operations.

## G  HEAD SPECIALIZATION ANALYSIS

In this section, we analyze Dynamic Mask Attention, revealing its unique advantages in handling long-range dependencies and dynamic context awareness.

We analyze the attention patterns learned by the model as shown in Figure 10, revealing how DMA creates content-aware sparse structures that adapt to different contextual needs. Unlike the uniform patterns of traditional attention mechanisms, each DMA attention head develops a unique sparse pattern: some heads focus on the most recent tokens to capture local context, while others attend to specific distant positions for long-range dependencies, and additional heads maintain broader context awareness for global understanding. This diversity allows the model to capture various types of dependencies simultaneously while maintaining computational efficiency, maximizing the utilization of each attention subspace.

---

[1]The implementation code for MHA is available at `https://github.com/Dao-AILab/flash-attention`.

[2]The implementation code for SWA is available at `https://github.com/Dao-AILab/flash-attention`.

[3]The implementation code for MLA is available at `https://github.com/deepseek-ai/FlashMLA`.

[4]The implementation code for NSA is available at `https://github.com/lucidrains/native-sparse-attention-pytorch`.

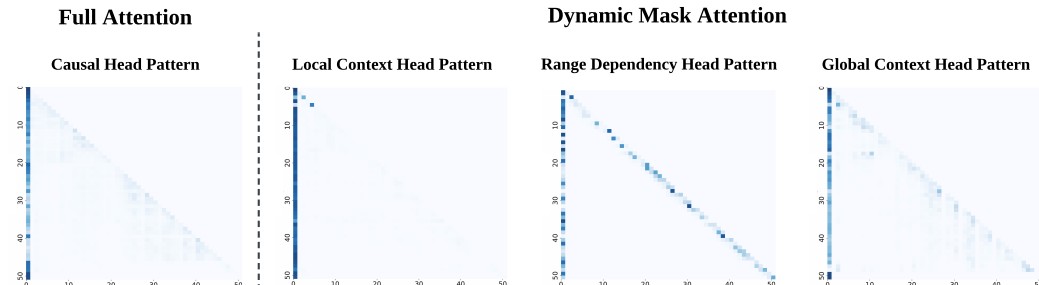

Figure 10: **Dynamic Mask Attention Heatmaps**. The heatmaps show the attention weights of each head in the Dynamic Mask Attention mechanism, indicating which tokens each head focuses on.

**Local Context Heads** tend to focus on tokens closest to the query, forming a local banded attention pattern. These heads are primarily responsible for capturing syntactic structures, phrase-level semantics, and local dependencies, which are particularly important for tasks requiring precise local context.

**Range Dependency Heads** exhibit the ability to attend to specific distant tokens. These heads are specialized for capturing long-range semantic associations. They can skip over large amounts of intermediate information, directly connecting parts that are far apart but semantically related, which is crucial for deep reasoning and contextual understanding.

**Global Context Heads** demonstrate a sparser but broader attention distribution, sampling key information from the entire sequence to form a holistic perception of the global context. These heads act like summarizers, responsible for integrating information from different parts to form a coherent global representation. This capability is crucial for tasks that require a comprehensive understanding of the entire input to make accurate predictions.

**Dynamic Adaptability.** The most significant advantage of DMA lies in its dynamism. These attention patterns are not static; they are dynamically generated based on the input content. This means the model can adjust its attention strategy in real-time, activating the most appropriate combination of heads when processing different tasks or text types. For example, when processing code, it might rely more on long-range dependency heads to track variable definitions and usages, whereas in a conversation, it might focus more on local context heads to understand the current exchange. This content-aware adaptability is the core advantage of DMA over static sparse attention methods.

This naturally occurring specialization is a direct result of the content-aware mask mechanism, enabling the model to effectively handle various types of dependencies while maintaining computational efficiency, achieving effective integration of multi-scale information. This hierarchical integration mechanism can effectively handle multi-level semantic structures in complex texts. It is worth noting that head specialization may also occur in traditional MHA, but the specialization patterns in DMA are more pronounced and functionally clearer, which may be a key reason for its superior performance across various tasks.

# H  DISCUSSION

In this section, we discuss the core deficiencies of existing sparse attention methods, analyze how Dynamic Mask Attention addresses these issues, and explore its limitations and future development directions.

## H.1  LIMITATIONS OF EXISTING APPROACHES

Existing sparse attention methods exhibit three critical deficiencies that limit their practical effectiveness:

**Post-hoc Sparsification Degradation.**  The performance degradation caused by post-hoc sparsification stems from the fundamental mismatch between existing methods and the optimization trajectory of pretrained models. As demonstrated by Chen et al. (Chen et al., 2024), retaining only the top 20% of attention weights covers only 70% of the total attention scores. This forced sparsification

strategy compels models to deviate from the optimal parameter configurations learned on large-scale corpora. More critically, this approach causes irreversible damage to key structural components in pretrained models, such as retrieval heads and copy heads, as these specialized attention heads are misidentified as "unimportant" and pruned during inference.

**Training-Inference Efficiency Gap.** Most sparse attention methods optimize only for inference, neglecting training-phase computational demands. This creates bottlenecks across LLM development: pretraining on long documents, long-context fine-tuning, and reinforcement learning. Without effective training-time sparsity support, these crucial phases remain constrained by $O(n^2)$ computational complexity, limiting development of more capable long-context models.

**Non-differentiable Components and Inefficient Backpropagation.** Non-differentiable components and inefficient backpropagation problems reveal the technical shortcomings of existing methods in terms of trainability. The discrete operations in methods like ClusterKV (Liu et al., 2024b) and MagicPIG (Chen et al., 2024) introduce discontinuities in computational graphs, which block gradient flow and hinder the learning of optimal sparse patterns. Even trainable methods like HashAttention (Desai et al., 2024) suffer from memory access inefficiencies due to token-granular selection, which is incompatible with the contiguous memory access and block-wise computation requirements of efficient attention techniques, such as FlashAttention. Consequently, these implementations are forced to revert to naive implementations with low hardware utilization, significantly degrading training efficiency.

## H.2 How Dynamic Mask Attention Addresses Core Issues

Dynamic Mask Attention systematically addresses the aforementioned fundamental issues through three core innovations, achieving unified, efficient, and sparse computation for both training and inference phases.

**Native Trainable Sparsity.** Native trainable sparsity is DMA's key innovation for addressing post-hoc sparsification issues. Unlike traditional methods, DMA embeds sparsity into the model architecture from the ground up, ensuring that sparse attention patterns are fully aligned with the model's optimization trajectory. Specifically, DMA retains complete, uncompressed KV caches $k = \text{concat}([k_1, \ldots, k_t])$ and $v = \text{concat}([v_1, \ldots, v_t])$, ensuring the original fidelity of historical information and precise recall capabilities, avoiding information bottlenecks that may arise from fixed-state compression in State Space Models. This comprehensive information retention mechanism enables DMA to precisely access any token in the historical context at any moment, without losing critical information due to lossy compression methods like Mamba. More importantly, DMA's sparsification occurs during the attention weight computation phase, rather than in post-training processing, ensuring that models do not deviate from pre-trained parameter configurations during sparsification, thereby protecting key structural components, such as retrieval heads and copy heads, from damage.

**Unified Training-Inference Architecture.** The unified training-inference architecture eliminates the fundamental gap in training-inference efficiency that exists in existing methods. DMA's dynamic weight computation $\delta = \exp(\tau(v\Delta) \times A)$ uses identical sparsification strategies during both training and inference phases. This consistency ensures that models can learn optimal sparse patterns during training and seamlessly apply these patterns during inference. This unified architecture particularly benefits three critical stages of modern LLM development: the pretraining stage can efficiently process long document sequences; the long-context fine-tuning stage can adapt to specific task requirements; the reinforcement learning stage can effectively update attention weights through policy gradients. DMA reduces computational complexity from $O(n^2)$ to $O(n \cdot w)$, enabling the training of larger-scale long-context models.

**Fully Differentiable Design.** The fully differentiable design ensures that DMA maintains gradient flow continuity throughout the entire computation process. The computation of dynamic mask weights $\delta$ is based entirely on differentiable operations, including linear transformations of value representations, non-negative activation functions $\tau(\cdot)$, and exponential functions, thereby avoiding

gradient interruptions caused by discrete operations such as k-means clustering and SimHash. Although the mask generation process involves topk operations, since it is not the core learning objective of DMA but merely a tool for sparse selection, we can maintain practical gradient propagation through continuous relaxation techniques or straight-through estimators. Moreover, the attention weight computation part is designed such that the gradients for masked positions should naturally be zero, so skipping computation and setting gradients to zero is the correct behavior. This design enables the model to learn optimal attention patterns that are sparse in an end-to-end manner, dynamically adjusting which historical positions are most critical for current reasoning, thereby achieving truly content-aware, selective computation. Additionally, each head in a multi-head attention mechanism can independently generate different sparse patterns, thereby maximizing the representational capabilities of the multi-head architecture by focusing on different information segments in distinct subspaces.

## H.3    Limitations and Future Works

Despite Dynamic Mask Attention's significant progress in addressing the core issues of existing methods, several limitations remain that warrant further exploration and improvement in future work.

**Adaptive Window Size Selection.**    Adaptive window size selection is the primary challenge facing DMA. While the current fixed window size design provides predictable computational complexity, it may not optimally adapt to the dynamic demands of different tasks and contexts. For instance, code generation tasks may require larger windows to capture long-range structural dependencies, while simple question-answering tasks may only need smaller windows. Future research directions include developing adaptive window size selection mechanisms based on task complexity, sequence length, and content features, potentially through reinforcement learning or meta-learning approaches to dynamically optimize window parameters. Alternatively, designing hierarchical multi-scale attention structures can be considered to capture dependencies across different ranges simultaneously.

**Position Encoding Enhancement.**    Our needle-in-a-haystack experiments revealed an intriguing phenomenon: trainable sparse attention mechanisms, such as DMA, exhibit stronger length extrapolation capabilities compared to dense attention when context lengths exceed the pretraining bounds. This finding suggests that the fundamental bottleneck for extrapolation may lie in the position encoding method rather than the attention mechanism itself. Current RoPE-based position encodings struggle with out-of-distribution sequence lengths, but DMA's dynamic sampling architecture offers a potential alternative pathway for encoding positional information. Specifically, the zero-order hold sampling values that are added as attention biases can be explored to explicitly incorporate positional information into these sampling values, potentially replacing or complementing RoPE to create a more extrapolation-friendly encoding scheme. Such an approach might leverage the inherent advantages of sparse attention's selective computation to create position representations that scale more naturally to unseen lengths. This direction could help address one of the most persistent challenges in long-context modeling: maintaining consistent positional understanding across arbitrary sequence lengths without requiring length-specific fine-tuning.

**Multi-Modal Extension.**    Multi-modal extension represents an essential direction for DMA development. The current DMA design is primarily optimized for text sequences; however, modern AI systems increasingly require processing mixed inputs of text, images, audio, and video. Attention sparsity in multi-modal scenarios exhibits more complex patterns: interactions between different modalities may require different attention distributions, temporally aligned multi-modal information may need synchronized attention mechanisms, and modality-specific long-range dependencies may require specialized sparse patterns. Future research can explore modality-aware dynamic mask generation, coordination mechanisms for cross-modal attention weights, and specialized sparse pattern designs for different modal characteristics.

