# OpenReview forum: "Dynamic Mask Attention: End-to-End Trainable Content-aware Sparse Attention"
_ICLR.cc/2026/Conference — Submitted to ICLR 2026_

### Official Review · Reviewer_TP8n · 2025-10-26

**Soundness:** 2
**Presentation:** 2
**Contribution:** 2
**Rating:** 2
**Confidence:** 4

**Summary:**

This paper proposes a sparse attention mechanism named Dynamic Mask Attention (DMA) to address the computational bottleneck in long-context modeling for Large Language Models. It generates content-driven dynamic masks based on value vectors and assigns continuous importance weights to tokens. This design avoids the non-differentiable hard binary masks used in previous methods and enables end-to-end training. Furthermore, the authors have developed efficient CUDA kernels to achieve practical acceleration. Experimental results demonstrate that DMA outperforms existing sparse attention baselines such as Native Sparse Attention (NSA) in terms of perplexity, downstream task accuracy, and long-sequence inference speed.

**Strengths:**

1. End-to-End Trainability: The primary contribution of this paper is a fully differentiable sparse attention mechanism. By generating masks from value vectors and adding them to the attention map, gradients can flow through the mask generation process. This allows the model to be optimized end-to-end.
2. Efficient Implementation: The authors not only propose a theoretical model but also provide an efficient implementation. It demonstrates significant training and inference speedups on long sequences.

**Weaknesses:**

1. Insufficient dynamism: Although the method is designed to be "content-aware" and "dynamic," the sparse pattern is only generated by values and the mask is broadcast to all queries, which limits its dynamism.
2. Unclear rationality of end-to-end training：The paper devotes considerable space to explaining why adding the obtained scores to the attention map achieves differentiability. However, differentiability is an obvious outcome, and the paper fails to explain why this approach enables the model to learn reasonable top-k sparse behavior.
3. Unclear decoding methodology：The paper uses the topk method for token selection, but does not clearly explain the model's behavior during the decoding stage.
4. Incomplete experiment：There is no comparison of training speeds, lacking inference speed comparisons with Flash Attention 2 and 3. Additionally, the model training was only conducted on 2k pre-training and subsequent 8k training, making it difficult to demonstrate scalability on longer samples. Moreover, ablation experiments on model components are missing.

**Questions:**

1. Why generate scores using the method in formula (1) instead of a simpler approach like using a linear projection followed by an activation function?
2. Regarding "Mask generation", "Mask Broadcasting", and formulas (2)(3), why are the top w tokens selected directly from the value scores of the complete sequence and broadcast to queries rather than being selected for each query individually?
3. What do the symbols in Table 2 of Appendix F mean? Why is this model configuration used for NSA?
4. What implementation is used for the speed in Figure 5, is it efficient implementations like flash attention and flash mla?
5. In the Native Trainable Sparsity paragraph of Appendix G.2, it is stated that the method proposed in this paper preserves the complete key-value cache and avoids information loss. However, the score for a particular token, which is calculated solely based on its value, does not change during the decoding process. Would this cause some tokens to never be activated, resulting in information loss?
6. Regarding the Unified Training-Inference Architecture paragraph in Appendix G.2, how is decoding specifically implemented? When encountering newly added key-value pairs, do all of them participate in the computation, or are the top w tokens reselected?
7. Have you attempted to conduct similar experiments based on GQA instead of MHA?

---

> ### Author Response · Authors · 2025-11-16
>
> We thank Reviewer for the insightful questions on dynamics, training mechanism, and decoding behavior. We clarify details of some problems you proposed in the public comment, where we also highlight the corresponding changes in Sections 3.2–3.4 in our revised PDF.
>
> 1. **On the Concern of Limited Dynamics**
> We agree that a purely Value-based global mask shared by all queries would be overly rigid. As summarized in Public Comment 5, we instead adopt a hybrid strategy of value-driven global $top-w_{kv}$ plus per-query dynamic slots $w_q$. The revision includes the corresponding formulas and clarifies why we avoid per-query top-k over the entire sequence: while more fully dynamic, it leads to unfavorable memory access patterns, weakens kernel fusion with Flash-style implementations, and significantly reduces the practical speed benefits. The final design aims to balance dynamics with implementation efficiency.
>
> 2. **On End-to-End Training and Learning Top-k Behavior**
> The revised text now provides a clearer account of how the model learns reasonable top-k behavior end-to-end (Public Comment 4; Section 3.2). During training, the Value-driven scores act as a differentiable bias added to the attention scores, and the forward pass uses a hard top-k mask; in the backward pass, only selected tokens participate in attention computation and thus receive larger gradients in the scoring network. Under task loss supervision, this creates a winner-take-most dynamic that aligns the learned sparsity with task objectives. We also compare this value-driven scoring with linear-projection-plus-activation approaches such as Retaining on long-context retrieval. Under the same 8K dense model and $w=4K$ extension to 16K, DMA achieves higher average RULER scores than Retaining while remaining very close to MHA (following Table), supporting our choice in terms of stability and effectiveness.
>
> | Method | SG1 | SG2 | SG3 | MK1 | MK2 | MK3 | MV | MQ | VT | CWE | FWE | QA1 | QA2 | Avg |
> |-|-|-|-|-|-|-|-|-|-|-|-|-|-|-|
> | MHA | 100.0 | 100.0 | 100.0 | 86.0 | 51.0 | 18.0 | 92.5 | 84.5 | 27.0 | 0.8 | 62.7 | 36.0 | 30.0 | 60.65 |
> | Retaining | 100.0 | 100.0 | 98.0 | 66.0 | 42.0 | 8.0 | 73.5 | 75.0 | 4.4 | 2.0 | 74.7 | 34.0 | 34.0 | 54.74 |
> | DMA | 100.0 | 100.0 | 98.0 | 70.0 | 44.0 | 14.0 | 93.0 | 82.5 | 35.6 | 2.6 | 74.7 | 36.0 | 36.0 | 60.49 |
>
> 3. **On Decoding Behavior and Unified Train–Inference Architecture**
> To avoid confusion about decoding, we now present a unified view of training and inference in the revision (Public Comment 5; Section 3.3). Both phases share the same sparsification rule; decoding differs only in its incremental update pattern. For new KV pairs during decoding, we compute their Value-based scores and merge them into the current global $top-w_{kv}$ set without re-scoring the entire history, and restrict QK computations to at most $w_{kv} + w_q$ positions per query. Evicted KV entries are moved from HBM to lower-tier memory to control VRAM usage, making it clear that decoding does not attend to the full history at every step but maintains the same sparsity principle as training.
>
> 4. **On Information Loss Risk and Stable Value Scores**
> We acknowledge the risk that fixed or slowly changing Value-based scores could prevent some tokens from re-entering the global set. As discussed in Public Comment 5, this is a conscious trade-off to obtain predictable memory usage and safe KV eviction. To mitigate potential information loss, we use per-query dynamic slots $w_q$ that allow the current query to recall a small number of tokens from the candidate pool, and we perform long-context continued training under the same sparsity rule so that the model learns to use this mechanism end-to-end. Empirically, DMA matches or outperforms MHA on RULER, LongBench, and other long-context tasks (Section 3.4; Table 2), suggesting that this trade-off does not lead to noticeable degradation in practice
>
> 5. **On Speed Implementations and FlashAttention**
> Speed-related concerns are addressed in more detail in Public Comment 6. Briefly, both training and inference rely on efficient open-source implementations; baselines include MHA, SWA, MLA, NSA, and other variants, with forward, backward, and decode times reported separately (Fig. 5; Appendix F, Table 5). For FlashAttention, we use FlashAttention2-based kernels where applicable on A100; as FlashAttention3 targets Hopper GPUs, we explicitly note that we do not include FA3 in our comparisons to avoid an unfair setting
>
> 6. **On Notation and NSA Configuration**
> Finally, following your suggestion and as mentioned in Public Comment 2, we extend the appendix with explanations for all symbols used in the configuration tables and describe how they correspond to the parameters in the original NSA paper. For NSA in particular, we strictly follow the default configuration from the original work, adjusting only global model size to match our parameter budget, which we document for reproducibility

---

> ### Author Response · Authors · 2025-11-25
>
> Dear Reviewer,
>
> This is a gentle reminder that the discussion period is concluding, and we hope you have had the chance to consider our responses.
>
> We also wanted to quickly point out that we have updated the revised manuscript PDF. We realized we forgot to highlight the changes in the previous version, and we apologize for that error. The new file now has all revisions clearly marked in blue, which should make it much easier to review.
>
> If you feel our responses and the revised manuscript have now satisfactorily addressed your concerns, it would be greatly appreciated if you could consider raising your score to reflect that the outstanding issues have been resolved.
>
> Thank you for your understanding and time!
>
> Best regards.

---

### Official Review · Reviewer_WvcK · 2025-10-27

**Soundness:** 3
**Presentation:** 2
**Contribution:** 3
**Rating:** 6
**Confidence:** 3

**Summary:**

This paper introduces Dynamic Masked Attention (DMA), a trainable, head-specialized, content-aware sparse attention mechanism. The key is that masks are generated by Value and importance weights are continuous, thus achieving expressiveness and full differentiability. Compared to the current SOTA, it reduces PPL and improves accuracy.

**Strengths:**

1. The idea of ​​using Value to generate importance weights is very innovative and can achieve better results than fullAttention.
2. The paper uses the head-wise specialization method and also analyzes the different focuses of each head. The experimental analysis here is quite sufficient.
3. The paper provides a detailed theoretical proof that this mask method can be trained.

**Weaknesses:**

1. In the experimental results section, the configurations of the comparison models are not detailed enough, so it is difficult to understand whether it is a completely fair comparison. For example, when comparing various sparse methods, the sparse ratio of each method need to be given.
2. The paper does not show the speed comparison with other sparse methods, which is also a very important evaluation indicator.

**Questions:**

1. Why do the full attention and NSA 8k needle indicators seem to be worse than other papers, such as the original NSA paper？what is the configuration of the full attention length extrapolation that seems to have a poor score?
2. In the speed comparison, it’s not fair to only compare PyTorch’s nativeScaled Dot-Product Attention. Can you compare it with flashAttn?
3. What is the block_size in the paper? Are there any ablation experiments with different block_size configurations?

---

> ### Author Response · Authors · 2025-11-16
>
> We thank Reviewer for the thoughtful comments on experimental design, speed evaluation, and configuration details. We clarify details of some problems you proposed in the public comment, where we also point to the new tables and figures in our revised PDF.
>
> 1. **On Configuration Details and Sparsity Ratios**
> We agree that the original submission did not provide enough configuration and sparsity details to fully assess fairness. As summarized in Public Comments 2 and 3, the revision adds configuration tables listing, for each method, the parameter count, number of training tokens, $n_{h_{kv}}$, context length, RoPE base, and other key hyperparameters, together with unified formulas for computational and memory complexity. Based on these formulas, we report effective sparsity ratios (active KV proportion) so that readers can see that DMA, NSA, and other sparse variants are compared under similar sparsity budgets. NSA’s configuration follows the default settings from the original paper, with only width/depth adjusted to match our target scale, as documented in Appendix F (Tables 3 and 4).
>
> 2. **On Speed Comparison and Missing Sparse Baselines**
> We fully agree that a comprehensive and fair speed comparison is crucial. In response, and as detailed in Public Comment 6, we now provide a systematic speed study on A100 with a representative configuration ($n_h=32$, $n_{h_{kv}}=8$, $d_h=128$), covering sequence lengths 256–32,768 for forward/backward and 256–524,288 for decode (Fig. 5). We compare DMA against MHA, SWA, MLA, and NSA, all using efficient open-source implementations (e.g., FlashAttention2 backends for MHA and SWA), and report forward, backward, and decode times separately; detailed configurations and implementation sources are given in Appendix F and Table 5. As noted in the overall response, FlashAttention3 targets Hopper GPUs, while our experiments are conducted on A100, so we do not include FA3 to avoid an unfair comparison. Within the A100 setting, the results show that DMA achieves substantial speedups over dense MHA at long sequence lengths and is competitive with or faster than other sparse baselines at similar sparsity levels.
>
> 3. **On Needle-in-a-Haystack Results vs. Other Papers**
> Your concern about cross-paper comparisons of needle-in-a-haystack results is well taken. As discussed in Public Comment 7, absolute scores on this benchmark depend on [model size, training data, and context configuration](https://arxiv.org/pdf/2404.06654), so results from our 1.7B model are not directly comparable to the 27B NSA experiments. We therefore avoid making cross-paper claims. Instead, we follow recent [standard practice](https://arxiv.org/pdf/2411.13676) for length extrapolation (e.g., pretraining at 2K, then extending context to 8K via [RoPE scaling](https://arxiv.org/pdf/2309.16039) and continued training) and compare DMA against the dense baseline under exactly the same pretraining and continued-training protocol. The revised manuscript describes this protocol in more detail (Section 3.4) so that the observed needle performance can be interpreted as a controlled comparison of attention mechanisms within our setup.

---

> ### Author Response · Authors · 2025-11-25
>
> Dear Reviewer,
>
> This is a gentle reminder that the discussion period is concluding, and we hope you have had the chance to consider our responses.
>
> We also wanted to quickly point out that we have updated the revised manuscript PDF. We realized we forgot to highlight the changes in the previous version, and we apologize for that error. The new file now has all revisions clearly marked in blue, which should make it much easier to review.
>
> If you feel our responses and the revised manuscript have now satisfactorily addressed your concerns, it would be greatly appreciated if you could consider raising your score to reflect that the outstanding issues have been resolved.
>
> Thank you for your understanding and time!
>
> Best regards.

---

### Official Review · Reviewer_3ohf · 2025-11-07

**Soundness:** 3
**Presentation:** 3
**Contribution:** 3
**Rating:** 6
**Confidence:** 3

**Summary:**

This paper proposes a method to train a language model with dynamic sparse attention. Concretely it uses the value vector for each attention head to generate a mask for token, which decides whether the token will be masked out for attention calculation.

Experiments show that training with the proposed method (DMA) achieve better better perplexity and downstream performance compared to dense attention and other sparse attention methods, such as NSA.

**Strengths:**

* The paper aims to improve the efficiency of language model by proposing a sparse attention method, which is well-motivated.
* Experiment are conducted on models from 80M to 1.7B scale, and demonstrate speed-up compared to dense attention, and performance improvement compared to the other sparse attention methods.

**Weaknesses:**

* Experiments are conducted on relatively small-scaled model (only up to 1.7B) and short context (trained only up to 8K sequence length). Experiments are larger scale model or longer context pre-training would be helpful to demonstrate more practical utility of the proposed method, for instance, LongBench contains context length > 8K.
* I am curious about the reason behind choosing MHA as a baseline (and implementation for DMA) instead of grouped query attention, which is a standard method to improve efficiency.

**Questions:**

* Regarding the hyper-parameter chosen for Table 2, could the authors explain on what basis are these numbers chosen? For instance, is this for fair comparison to the tokens attended, FLOPs, or latency?
* As the mask is depended on the value vectors, will the mask pattern change during generation or will the pattern be relatively stable? It will also be helpful to report per-task performance on LongBench and [RULER](https://arxiv.org/abs/2404.06654) to conduct a more fine-grained evaluation on different types of task that process long input.
* There are several typos in the paper that makes the sentence unreadable: line 16-18 and line 24 in the abstract.

---

> ### Author Response · Authors · 2025-11-16
>
> We thank Reviewer for the careful review and helpful suggestions. We clarify details of some problems you proposed in the public comment, where we also indicate the corresponding changes in the main text and appendix in our revised PDF.
>
> 1. **On Model Scale and Context Length**
> We agree that pretraining larger models with longer context would further demonstrate the practical value of the proposed method. As summarized in Public Comment 1, we focus on a 1.7B model and 2K–8K context to balance compute cost and reproducibility, which is a common scale in [recent long-context work](https://arxiv.org/pdf/2411.13676). In the revision, we additionally perform continued training of the same 1.7B model up to 16K context under a unified DMA sparsification strategy, and evaluate on long-context benchmarks such as [LongBench](https://arxiv.org/abs/2308.14508) and [RULER](https://arxiv.org/abs/2404.06654) (Overall Item 1; Section 3.4 and Table 2). These results show that DMA maintains strong performance when moving from 8K to 16K context while providing significant speedups.
>
> 2. **On MHA/GQA as the Baseline**
> As clarified in Public Comment 3, all our main experiments actually use grouped-query attention (GQA) with $n_{h_{kv}} = n_h / 2$, and DMA is implemented on top of exactly the same GQA configuration. The original use of the term “multi-head attention” may have introduced ambiguity; we have now made the GQA setting explicit in both the method and experimental setup sections, and we list $n_{h_{kv}}$ in the configuration tables (Appendix F, Table 3). This makes it clear that our comparisons are against a strong and hardware-efficient baseline rather than a weaker dense-attention setting.
>
> 3. **On Table 2 Hyperparameters and Fairness**
> The goal of Table 2 is to compare different attention variants under aligned parameter counts, training tokens, and effective sparsity. As detailed in Public Comment 2 and Appendix F (Tables 3 and 4), we now provide configuration tables and unified complexity formulas for all methods, and report the resulting sparsity ratios (active KV proportion). For NSA, MLA and other variants, we follow the recommended settings from the original papers and only adjust global width/depth to match our target parameter scale, which is explicitly documented. These additions make the fairness of the comparisons clearer and easier to verify.
>
> 4. **On Mask Stability, Dynamics, and Long-Context Evaluation**
> The concern about mask dynamics and long-context behavior is discussed in Public Comments 4 and 5. In short, the Value-based scores used to construct the global dynamic mask are generally stable across training and generation, enabling safe KV eviction and reduced memory, while a small number of per-query dynamic slots $w_q$ reintroduce query-dependent flexibility on top of the global mask. We present the update rules and complexity analysis in the revised method section (Section 3.3, lines 236–304). For long-context evaluation, we start from an 8K pretrained model, extend the sequence length to 16K with a total budget $w = 4\mathrm{K}$, and report detailed RULER scores (Public Comment 4; Table 2), which show that the learned dynamic masks remain effective when extrapolating to longer contexts.
>
> 5. **On Typos and Grammar**
> We appreciate the pointer to typos and awkward phrases. Following your suggestion (Public Comment 7), we have carefully revised the abstract around lines 16–18 and 24 and simplified the corresponding sentences to improve readability. We also cleaned up several other minor wording issues in the main text and appendix, and ensured that notation and terminology are consistent throughout the paper.

---

> ### Author Response · Authors · 2025-11-25
>
> Dear Reviewer,
>
> This is a gentle reminder that the discussion period is concluding, and we hope you have had the chance to consider our responses.
>
> We also wanted to quickly point out that we have updated the revised manuscript PDF. We realized we forgot to highlight the changes in the previous version, and we apologize for that error. The new file now has all revisions clearly marked in blue, which should make it much easier to review.
>
> If you feel our responses and the revised manuscript have now satisfactorily addressed your concerns, it would be greatly appreciated if you could consider raising your score to reflect that the outstanding issues have been resolved.
>
> Thank you for your understanding and time!
>
> Best regards.

---

### Author Response · Authors · 2025-11-16

We thank the reviewers for their constructive feedback. In response to the main concerns on model scale and long-context setting, configurations, GQA setup, value-driven scoring and dynamics, decoding strategy, and speed comparison, we have made the following additions and clarifications in the revised manuscript:

1. Model Scale and Long-Context Setup
Our main experiments use a 1.7B-parameter model with 2K–8K context, which we found to be a reasonable trade-off between compute cost and reproducibility for long-context pretraining and method validation. In the revision, we additionally perform 16K long-context continued training on the same 1.7B model, and report detailed results on LongBench and RULER to better demonstrate the behavior of our method under longer inputs (see Section 3.4 lines 507-516 and Table 2).

2. Configurations
We add complete configuration tables and complexity formulas for all attention variants in the appendix. For NSA, MLA and other baselines, we follow the recommended configurations from their original papers; for NSA in particular, we further align with the default configuration in the original work and explicitly explain all symbols used in the tables. Overall, we match total parameter counts, training budgets, and effective sparsity levels across methods as closely as possible to ensure fair comparisons (see Appendix F, Tables 3 and 4).

3. GQA Setup
All main experiments are conducted with a GQA architecture, using a unified setting of $n_{h_{kv}} = n_h / 2$, and DMA is implemented on top of the same GQA configuration. We now clearly state and emphasize this in the methodology and experimental setup sections so that readers can accurately reproduce our results and see that our setting is aligned with standard efficient attention configurations (see Appendix F and Table 3).

4. Value-Driven Scoring and Top-k Behavior
During training, we treat the differentiable scores derived from the Value as an additive bias to the attention scores: the forward pass applies a hard top-k mask, and the backward pass propagates gradients only through the selected tokens. Under the supervision of the task loss, useful tokens receive larger gradients and are more likely to be selected again, which leads to a reasonable learned sparsity pattern.
We also compare value-driven scoring with linear-projection-plus-activation approaches such as Retaining. Under the same 8K dense model, continued training to 16K with w=4K, DMA achieves a higher average RULER score than Retaining while remaining very close to MHA, supporting the effectiveness and stability of our design:

|Method|Avg|
|-|-|
|MHA|60.6|
|Retaining|54.7|
|DMA|60.5|

5. Dynamics and Decoding Strategy
We now provide a unified formal description of both training and decoding. At decoding time, we adopt a hybrid strategy of value-driven global $top_{w_{kv}}$ + per-query dynamic slots $w_q$:
- Each attention head maintains a global KV index set of size at most $w_{kv}$, updated incrementally by top-k over Value scores. KV entries evicted from this set are moved to slower memory, keeping the active KV cache size per head constant for arbitrarily long sequences.
- For each query, we additionally select up to $w_q$ query-aware positions from a candidate subset. The final attention index set is the union of the global and per-query parts, with size at most $w_{kv}+w_q$, reducing complexity from $O(n_htd_h)$ to $O(n_h(w_{kv} + w_q)d_h)$.
- The global sparsity pattern remains relatively stable over time, while $w_q$ provides local dynamics for the current query.
Regarding the concern that some tokens may remain inactive for a long time, this is a deliberate trade-off to enable memory-friendly KV eviction. Through the candidate set design and the per-query dynamic slots $w_q$, important tokens still have opportunities to be recalled. As shown on long-context benchmarks such as RULER (see Table 3), this trade-off does not lead to noticeable performance degradation in practice.

6. Speed and Training Efficiency
We conduct systematic speed evaluations on A100 with a representative configuration, covering sequence lengths 256–32K for forward/backward and 256–524K for decode (see Fig. 5).
We compare against MHA, SWA, MLA, NSA and other variants, all implemented with efficient open-source kernels; detailed configurations and implementation sources are provided in Appendix F and Table 5. For training, we report forward and backward times separately.
The results show that DMA is significantly faster than MHA and other sparse baselines on long sequences, and achieves similar efficiency to SWA.

7. Long-Context Benchmarks and Multi-Task Results
After 16K long-context continued training, we evaluate the model on multiple benchmarks including LongBench, and RULER, and directly compare MHA, NSA, and DMA (see Table 2). The results show that, while providing substantial speedups, DMA matches or slightly outperforms MHA on most tasks.

---

### Author Response · Authors · 2025-11-25

Dear Reviewers,

We sincerely appreciate your time and effort in reviewing our manuscript and offering valuable suggestions. As the author-reviewer discussion phase has concluded, we are writing to follow up on our responses.

We would also like to make a quick clarification regarding the revised manuscript. We noticed that in the PDF we initially uploaded, we unfortunately forgot to highlight the changes. We sincerely apologize for this oversight. We have now uploaded a new version of the revised manuscript with all changes marked in blue for your convenience.

We hope that our detailed responses, together with this newly highlighted version, have effectively addressed your concerns. If you require any further clarification or have additional feedback on the revisions, please do not hesitate to let us know. We are more than willing to continue our communication.

Thank you again for your consideration.

Best regards.

---

### Author Response · Authors · 2025-12-01

Dear Area Chair,

Thank you for taking the time to review our submission. Given that none of the three reviewers have responded during the discussion phase, we would like to briefly summarize the additional experiments and revisions we have completed in response to the reviewers' core concerns, to help you efficiently understand the current status of the paper.

The current status of the three reviewers is as follows (none have replied):

Reviewer | Original | Current Status
--- | --- | ---
3ohf | 6 | 6 (Not Reply)
WvcK | 6 | 6 (Not Reply)
TP8n | 2 | 2 (Not Reply)

Summary of Improvements Addressing Core Concerns

**Reviewer 3ohf**: Original Score 6. This reviewer mainly focused on model scale and long-context settings (only 8K), baseline selection (MHA vs GQA), and the dynamic stability of the mask.
We addressed these comprehensively by:
1. **Extending Long-Context Experiments**: We performed 16K long-context continued training on the 1.7B model and added detailed results on LongBench and RULER (Table 2), demonstrating the method's effectiveness on longer sequences.
2. **Clarifying GQA Setup**: We clarified that all experiments are based on GQA ($n_{h_{kv}} = n_h / 2$), and DMA is implemented on top of this, ensuring a fair comparison with efficient baselines.
3. **Refining Configurations and Analysis**: We added detailed configuration tables and complexity formulas, and explained the design rationale of the hybrid dynamic strategy (Global Top-k + Local Dynamic Slots) to address concerns about mask stability.

**Reviewer WvcK**: Original Score 6. This reviewer mainly focused on the fairness of experimental configurations (e.g., sparsity ratio details), inference speed comparison with other sparse methods, and cross-comparison issues with Needle-in-a-Haystack results.
We addressed these comprehensively by:
1. **Adding Detailed Configurations and Sparsity Ratios**: We added complete configuration tables in the appendix (Tables 3 and 4), including parameter counts, training tokens, and effective sparsity ratios, ensuring transparency and fairness in comparisons.
2. **Systematic Speed Evaluation**: We added a systematic speed study on A100 (Fig. 5), comparing forward, backward, and decoding speeds of MHA, SWA, MLA, NSA, and other baselines across different sequence lengths, demonstrating DMA's significant speed advantage on long sequences.
3. **Clarifying Experimental Protocol**: We explained the controlled variable settings for the Needle-in-a-Haystack experiments and clarified why absolute scores should not be directly compared across models with different scales or training data.

**Reviewer TP8n**: Original Score 2. This reviewer mainly questioned the insufficient dynamism of the method (Global Mask), the rationality of end-to-end training for learning the Top-k mechanism, the clarity of the decoding strategy, and the completeness of experiments.
We addressed these comprehensively by:
1. **Elaborating on Hybrid Dynamic Mechanism**: We detailed the hybrid strategy of "Value-based Global Top-k + Per-Query Dynamic Slots $w_q$", clarifying how the model balances dynamism with hardware efficiency.
2. **Explaining Training Mechanism and Effectiveness**: We added comparative experiments with linear-projection-plus-activation approaches (e.g., Retaining) on long-context retrieval tasks (Table in response), verifying the effectiveness of our Value-driven scoring mechanism in learning sparsity patterns.
3. **Unifying Training-Inference Description**: We rewrote the method section to provide a unified formal description of training and decoding, explaining how new KV pairs are handled during decoding while maintaining the same sparsity principle.

---

### Meta-Review · Area_Chair_Hakn · 2026-01-10

**Summary:**

1. Reviewer TP8n and the SAC note that the sparse pattern generated by the method is based solely on value tensors, which limits its dynamism. The authors addressed this to some degree by revising the method to optionally include "query-dependent dynamic slots" -- detailed in 2.3, a new section of the paper. While it is nice to see this revision, it would have been better to see this baked in from the very beginning and a more coherent part of the narrative. Furthermore, as far as I can tell, there are no experiments that use this query-dependent slots.
2. There are no details about the efficient CUDA kernel implementation.
3. There are minor typos in the paper:
- e.g. SWA (Beltagy et al., 2020) in line 371 (wrong citation)
- unlined -> underlined in line 445
4. Models larger than 1.7B (e.g. 7B) and longer sequence lengths should be used, given that the paper focus is on efficient attention, as noted by Reviewer 3ohf. The authors did add results for a longer sequence length (8k -> 16k) but no updates to the model size. Results on a 7B would be very nice to see, along with a different architecture (not just Qwen3).
5. The authors pre-train from scratch, training the parameters for DMA alongside the rest of the Transformer. What would happen if they took an open source model checkpoint, trained with full attention, and then continued pre-training with DMA?
6. The paper packs a lot of information and the paper's readability, while not terrible, has room for improvement.
7.  Was the KV cache used for DMA and baselines the same size? Was it unbounded or bounded? I wasn't able to find find these in the paper.

Overall, while I commend the authors for the substantial work in the the rebuttal, concerns remain and I do think the overall flow and readability of the paper could be improved. Thus, I lean to reject but would consider accept if someone champions the paper.

**Reviewer Concerns:**

Most of the reviewers concerns were touched upon in the rebuttal.

**Reviewer Scores:**

While the authors' rebuttal was fairly substantial and included additional experiments and paper changes, I believe its effect on reviewer scores would be as follows:

* Reviewer 3ohf: unlikely to change their score from 6 (lukewarm positive)
* Reviewer WvcK: also unlikely to change their score from 6.
* Reviewer TP8n: would increase their score from 2 to 3/4, but remain a reject on the paper.

---

### Decision · Program_Chairs · 2026-01-26

Reject